# Loss of p53 suppresses replication-stress-induced DNA breakage in G1/S checkpoint deficient cells

Bente Benedict[1†], Tanja van Harn[1†], Marleen Dekker[1], Simone Hermsen[1], Asli Kucukosmanoglu[1], Wietske Pieters[1], Elly Delzenne-Goette[1], Josephine C Dorsman[2], Eva Petermann[3], Floris Foijer[1,4], Hein te Riele[1]*

[1]Division of Tumor Biology and Immunology, The Netherlands Cancer Institute, Amsterdam, The Netherlands; [2]Department of Clinical Genetics, VU University Medical Center, Amsterdam, The Netherlands; [3]School of Cancer Sciences, University of Birmingham, Birmingham, United Kingdom; [4]European Research Institute for the Biology of Ageing, University Medical Center Groningen, Amsterdam, The Netherlands

**Abstract** In cancer cells, loss of G1/S control is often accompanied by p53 pathway inactivation, the latter usually rationalized as a necessity for suppressing cell cycle arrest and apoptosis. However, we found an unanticipated effect of p53 loss in mouse and human G1-checkpoint-deficient cells: reduction of DNA damage. We show that abrogation of the G1/S-checkpoint allowed cells to enter S-phase under growth-restricting conditions at the expense of severe replication stress manifesting as decelerated DNA replication, reduced origin firing and accumulation of DNA double-strand breaks. In this system, loss of p53 allowed mitogen-independent proliferation, not by suppressing apoptosis, but rather by restoring origin firing and reducing DNA breakage. Loss of G1/S control also caused DNA damage and activation of p53 in an *in vivo* retinoblastoma model. Moreover, in a teratoma model, loss of p53 reduced DNA breakage. Thus, loss of p53 may promote growth of incipient cancer cells by reducing replication-stress-induced DNA damage.
DOI: https://doi.org/10.7554/eLife.37868.001

*For correspondence:
h.t.riele@nki.nl

†These authors contributed equally to this work

Competing interests: The authors declare that no competing interests exist.

## Introduction

To prevent cells become cancerous, different cell-cycle checkpoints can be activated to halt cell cycle progression. The G1/S checkpoint is responsible for controlling S phase entry and key effectors of this checkpoint are the retinoblastoma (Rb) proteins pRB, p107 and p130. Anti-proliferative conditions, such as lack of growth factors, suppress the activity of the D-type cyclin-dependent kinases (CDKs) CDK4 and CDK6. This results in hypo-phosphorylation of the Rb proteins, which can then bind E2F transcription factors thereby inhibiting the transcription of E2F target genes required for S-phase entry (*Bertoli et al., 2013*; *Burkhart and Sage, 2008*). In a majority of human tumors, the G1/S checkpoint is lost, for example by loss of pRB or the CDK inhibitor p16[INK4A], or by overexpression of Cyclin D1 (*Ho and Dowdy, 2002*; *Weinberg, 2007*) and insensitivity to antigrowth signals is an hallmark of tumor cells (*Hanahan and Weinberg, 2000*). Cells lacking the G1/S phase checkpoint can start synthesizing DNA under non-permissive conditions which may lead to DNA damage.

To deal with DNA damage, cells have evolved another cell cycle checkpoint that is part of the DNA-damage response (DDR) (*Jackson and Bartek, 2009*). Activation of the DNA damage checkpoint triggers cellular senescence or cell death, thereby providing an intrinsic biological barrier against tumor progression (*Bartkova et al., 2006*; *Gorgoulis et al., 2005*). It is often rationalized

**eLife digest** Healthy cells go through a strictly regulated process called the cell cycle in order to divide. During this cycle the cell's DNA is duplicated and the two copies are equally distributed between the two newly formed cells. Duplicating DNA is a complex procedure that can go wrong and damage the DNA. This damage, in turn, can cause cells to stop growing or even die.

Normal cells only start replicating their DNA when there are substances known as growth factors in the environment. Without growth factors cells remain in the first phase of the cell cycle, known as G1. Most cancer cells, however, lack this 'G1 checkpoint' and enter the cell cycle even when growth factors are absent. This leads to DNA replication problems and damage that should cause the cells to die. Yet a characteristic of cancer cells is that they overcome these problems to grow and divide uncontrollably.

Cancer cells also often lack a protein called p53. Previous studies demonstrated that the lack of p53 helps tumor cells to survive by maintaining cell growth and reducing the likelihood of cell death. By growing cells in culture without growth factors, Benedict, van Harn et al. now show that p53 also helps cells that lack the G1 checkpoint to continue dividing.

In the experiments, cells that lacked the G1 checkpoint but still contained the p53 protein suffered from DNA replication problems and DNA damage, and subsequently died. Deleting p53 from these cells stimulated DNA replication, stopped cells from dying and helped to prevent the DNA from getting damaged. Cells could thus grow and proliferate under unfavorable conditions. Benedict, van Harn et al. also deleted p53 in tumor cells growing under the skin of mice and observed less DNA damage in these cells than in tumor cells that still have p53.

Despite reduced levels of DNA damage, the cells still had severe DNA replication problems. It is possible that these cells rely on mechanisms that allow just enough DNA replication to occur to support their proliferation. Cancer cells may therefore be highly vulnerable to drugs that interfere with these mechanisms, since they are already using them as a last resort. Future experiments will be needed to identify these mechanisms.

DOI: https://doi.org/10.7554/eLife.37868.002

that inactivation of *Trp53*, a central player in the DDR and the most frequently mutated gene in human cancer (*Olivier et al., 2010*), promotes tumorigenesis by counteracting apoptosis and senescence induced by a defective G1/S checkpoint (*Sherr and McCormick, 2002*; *Reinhardt and Schumacher, 2012*; *Polager and Ginsberg, 2009*; *Bunz et al., 1998*; *Bieging et al., 2014*). However, here we present an unanticipated effect of p53 loss in cells that lack G1/S control.

To study the consequences of G1/S checkpoint loss in a well-defined system, we used primary mouse embryonic fibroblasts (MEFs) in which the three retinoblastoma (Rb) genes were inactivated. Previously, we and others demonstrated that these so-called triple knockout (TKO) MEFs can enter S phase without mitogenic signaling (*Dannenberg et al., 2000*; *Sage et al., 2000*). However, proliferation of TKO MEFs was still mitogen dependent: without mitogens, most cells became apoptotic whereas surviving cells arrested in a G2-like state. Suppression of apoptosis by ectopic expression of Bcl2 (TKO-Bcl2 MEFs) revealed that G2 arrest resulted from induction of p27$^{Kip1}$ and p21$^{Cip1}$ that inhibit Cyclin A- and B1-dependent kinase activity (*Foijer et al., 2005*). Induction of p21$^{Cip1}$ upon mitogen deprivation may be indicative for DNA damage (*Karimian et al., 2016*). Intriguingly, we previously showed that RNAi-mediated suppression of p53 and thereby reduction of p21$^{Cip1}$ levels revitalized CDK activity and supported mitogen-independent proliferation of Rb-protein-deficient cells (*Foijer et al., 2005*). In the present study, we provide mechanistic insight into the relief of proliferative arrest in mitogen-deprived TKO cells by p53 loss. We show that the DNA DSBs observed in mouse and human cells lacking G1/S phase control are caused by replication stress reflected by decreased replication speed and reduced origin firing. Inactivation of *p53* allowed for mitogen-independent proliferation, not only by suppressing apoptosis but also by restoring the levels of origin firing and reducing DSB formation. Similarly, in an *in vivo* model and in Rb-protein-deficient human cells, DNA breakage was reduced by loss of *p53*.

## Results

### Loss of p53/p21$^{Cip1}$ allows mitogen-independent proliferation of cells lacking the G1/S checkpoint

Consistent with our previous observations (*Foijer et al., 2005*), mouse embryonic fibroblast (MEFs) lacking the three retinoblastoma proteins and overexpressing the anti-apoptotic gene *Bcl2* (TKO-Bcl2 MEFs) ceased proliferation upon mitogen deprivation (*Figure 1A*, black line) and arrested in a G2-like state (*Figure 1C*, upper panel). We also reported that proliferation was rescued by RNAi-mediated knockdown of *Trp53*, the gene that encodes the p53 protein (TKO-p53RNAi MEFs) (*Foijer et al., 2005*). However, in recent experiments, proliferation of mitogen-starved TKO-p53RNAi MEFs appeared transient and was followed by severe cell loss (*Figure 1A*, green line), possibly as a result of residual p53 activity (*Figure 1—figure supplement 1A*). We therefore exploited CRISPR/Cas9 technology to create full *Trp53* knockout (KO) TKO MEFs (*Figure 1—figure supplement 1A*). Disruption of *p53* clearly rescued proliferation of mitogen-starved TKO MEFs (TKO-p53KO) and this effect was even greater in TKO MEFs expressing Bcl2 (TKO-Bcl2-p53KO), which reached 100% confluency (*Figure 1A*, blue and red lines). The improved proliferative capacity was accompanied by reduced apoptosis (*Figure 1B*) and the absence of G2 arrest (*Figure 1C*, lower panel, *Figure 1—figure supplement 1B*). Mitogen-deprived TKO-Bcl2-p53KO cells maintained a cell cycle profile similar to cells cultured in the presence of mitogens (*Figure 1C*, lower panel) and, unlike TKO-Bcl2 cells, continued to incorporate high levels of nucleotides (*Figure 1D*).

Not only loss of *p53*, but also disruption of its downstream target *Cdkn1a*, the gene that encodes the p21$^{Cip1}$ protein (*Figure 1—figure supplement 1C*), rescued proliferation of mitogen-deprived TKO-Bcl2 cells (*Figure 1E*). Apparently, the induction of p21$^{Cip1}$, which we previously found to inhibit Cyclin A- and B1-dependent kinases (*Foijer et al., 2005*), was critical for G2-like arrest of mitogen-deprived TKO cells. The p53/p21$^{Cip1}$ axis is part of the DNA damage response (DDR) and its activation is consistent with the high levels of DNA double-strand breaks (DSBs) that accumulated in arrested TKO-Bcl2 cells (*van Harn et al., 2010*). To understand how disruption of p53/p21$^{Cip1}$ rescued proliferation, we investigated the mechanism of DSB formation.

### Mitogen deprivation causes S-phase delay

We studied cell cycle progression of individual cells using the Fucci system, in which fluorescent proteins fused to the degradation motifs of Cdt1 and Geminin mark G1 and S/G2 cells, respectively (*Sakaue-Sawano et al., 2008*). In the presence of mitogens, TKO-Bcl2 and TKO-Bcl2-p53KO MEFs proliferated with a cell-cycle duration of 10 to 15 hr (*Figure 2A,B*, left). In the absence of mitogens, TKO-Bcl2 MEFs arrested in S/G2 phase, either immediately or after one cell cycle (*Figure 2A*, right). In contrast, mitogen-deprived TKO-Bcl2-p53KO MEFs could be followed for two or three cell divisions (*Figure 2B*, right), although G1 and S/G2 phase durations were increased, together encompassing 15 to 30 hr. These tracking experiments confirm that TKO-Bcl2-p53KO MEFs can proliferate in the absence of mitogens albeit at slower pace.

### p53/p21$^{Cip1}$ knockout suppresses DSBs formation

Cell cycle delay may be caused by DSBs that accumulate in mitogen-deprived TKO-Bcl2 MEFs (*van Harn et al., 2010*). This level was comparable to irradiation with 20 Gy, which is expected to severely impair mitosis resulting in cell death (*Zachos et al., 2003*). Nonetheless, TKO-Bcl2-p53KO and TKO-Bcl2-p21KO MEFs were able to proliferate mitogen-independently. We therefore investigated whether *Trp53* or *Cdkn1a* inactivation affected DSB formation as a consequence of mitogen deprivation by performing neutral comet assays (*Olive and Banáth, 2006*). Mitogen restriction of TKO-Bcl2 MEFs caused a clear increase in tail moment, an indicator of the level of DSBs (*Figure 3A, B*). In contrast, the tail moments in TKO-Bcl2-p53KO and TKO-Bcl2-p21KO MEFs were not significantly increased by mitogen depletion (*Figure 3B*) although the basal levels of DSBs (*i.e.*, in the presence of mitogens) were somewhat higher compared to TKO-Bcl2 cells. Possibly, MEFs accumulated some DNA damage under optimal culture conditions that was tolerated or not adequately repaired in the absence of p53/p21$^{Cip1}$ activity (*Levine and Oren, 2009*; *Williams and Schumacher, 2016*). Nevertheless, the critical observation here is that the induction of DNA breakage *due to mitogen deprivation* was suppressed in the absence of p53/p21$^{Cip1}$.

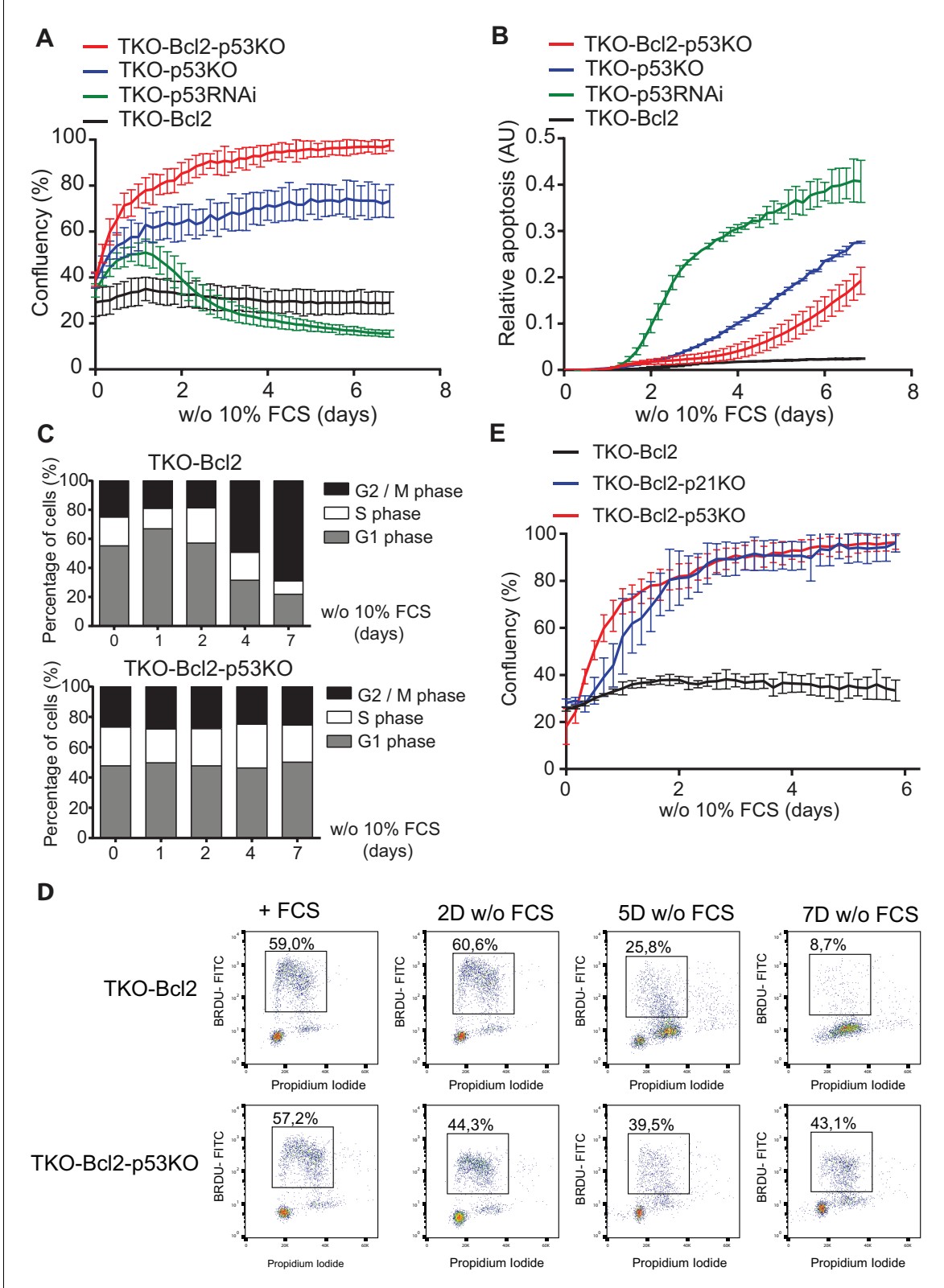

**Figure 1.** Loss of p21/p21$^{Cip1}$ promotes proliferation of mitogen-deprived MEFs lacking G1/S phase checkpoint. (**A**) IncuCyte growth curves of TKO-Bcl2 (black), TKO-p53RNAi (green), TKO-p53KO (blue) and TKO-Bcl2-p53KO (red) MEFs in the absence of 10% FCS. (**B**) Apoptosis levels of TKO-Bcl2 (black), TKO-p53RNAi (green), TKO-p53KO (blue) and TKO-Bcl2-p53KO (red) MEFs in the absence of 10% FCS. Apoptosis was measured by fluorescent signal upon caspase three cleavage and normalized to cell confluency. (**C**) Cell cycle distribution based on propidium iodide content of TKO-Bcl2 MEFs

*Figure 1 continued on next page*

Figure 1 continued

(upper panel) and TKO-Bcl2-p53KO MEFs (lower panel) in the absence of 10% FCS for the indicated days. (**D**) BrdU flow cytometry analysis of the cell cycle distribution of TKO-Bcl2 and TKO-Bcl2-p53KO MEFs in the absence of 10% FCS for the indicated days. Percentage of BrdU-labeled cells is indicated. (**E**) IncuCyte growth curves of TKO-Bcl2 (black), TKO-Bcl2-p53KO (red) and TKO-Bcl2-p21KO (blue) MEFs in the absence of 10% FCS. Experiments in A, B and E were performed in triplicate. Error bars show standard deviation (sd).
DOI: https://doi.org/10.7554/eLife.37868.003

The following figure supplement is available for figure 1:

**Figure supplement 1.** Reduced G2 arrest in mitogen-starved TKO-p53RNAi and TKO-p53KO MEFs.
DOI: https://doi.org/10.7554/eLife.37868.004

It is known that p53 modulates different DNA repair pathways (*Bieging et al., 2014*; *Williams and Schumacher, 2016*). Could the level of DSBs in mitogen-deprived p53KO MEFs be reduced by passage through M phase and subsequent repair in G1? To examine this possibility, we blocked cell cycle progression towards G1 by culturing cells in medium without mitogens, but containing nocodazole (*Figure 3—figure supplement 1*). This allowed us to measure the level of DSBs in TKO-Bcl2 and TKO-Bcl2-p53KO cells in comparable cell cycle phases, between S and M phase. In the presence of nocodazole, the same results were obtained: mitogen-deprived TKO-Bcl2 MEFs showed the expected increase in tail moment, while the tail moments of TKO-Bcl2-p53KO MEFs were still not increased (*Figure 3C*).

To directly investigate whether p53 status affected repair of replication-stress-induced DSBs, we treated *mitogen-stimulated* TKO-Bcl2 MEFs with 2 mM hydroxyurea (HU) for 1 hr in order to induce and alleviate replication stress instantaneously. HU depletes the cells of nucleotides, which results in stalling and collapsing of replication forks and hence DNA breakage (*Bianchi et al., 1986*; *Koç et al., 2004*). When comparing cells harvested immediately after HU treatment and cells harvested 30 min after HU treatment, we observed an equally strong decrease in tail moment in TKO-Bcl2 and in TKO-Bcl2-p53KO MEFs (*Figure 3D*). This indicates that the repair of DSBs induced by HU treatment was independent of p53 status. Assuming that repair of replication-stress-induced DSBs under mitogen-deprived conditions follows similar rules, these results suggest that reduced levels of DSBs in mitogen-deprived TKO-Bcl2-p53KO cells resulted from suppressed formation rather than increased repair of DSBs.

## Mitogen-deprived TKO-Bcl2 MEFs suffer from replication stress

To study the mechanism of DNA breakage, we assessed the quality of DNA replication in mitogen-deprived TKO-Bcl2 MEFs by looking at co-localization of the thymidine analogue chloro-deoxyuridine (CldU, marking DNA replication) and γ-H2AX (marking DNA damage). While the number of cells containing CldU foci gradually decreased in mitogen-starved TKO-Bcl2 MEFs, virtually all CldU foci that were still present after 4 and 7 days co-localized with γ-H2AX foci (*Figure 4A,B*). Furthermore, the gradual increase of phosphorylated Chk1 (pChk1), a target of ataxia telangiectasia related (ATR), is indicative for accumulation of single-stranded DNA (*Figure 4C*). Taken together, these results are indicative for perturbed replication in mitogen-deprived TKO-Bcl2 MEFs.

We next visualized the progression of individual replication forks using a DNA fiber assay (*Tuduri et al., 2010*). Sequential pulse-labeling of newly synthesized DNA strands with the thymidine analogs CldU (red tracks) and iodo-deoxyuridine (IdU, green tracks) identifies ongoing replication forks and new origin firing (*Figure 5A*). The length of double-labelled tracks in TKO-Bcl2 MEFs cultured with FCS indicated an average fork speed of 1.66 kb/min (*Figure 5B*). In the absence of *p53* the average fork speed was somewhat lower, 1.37 kb/min, consistent with a previous study (*Klusmann et al., 2016*). Mitogen deprivation caused a progressive decline in replication speed, somewhat unexpectedly not only in arresting TKO-Bcl2 MEFs but also in proliferating TKO-Bcl2-p53KO MEFs (*Figure 5B*). Prolonged S-phase and decelerated DNA synthesis indicate that mitogen-deprived TKO-Bcl2-p53KO MEFs were able to proliferate despite sustained replication stress.

## Nucleotide deficiency contributes to perturbed DNA replication

Disruption of the nucleotide pool can contribute to replication stress (*Bester et al., 2011*; *Poli et al., 2012*) and may therefore be the underlying cause of reduced replication speed, DSB

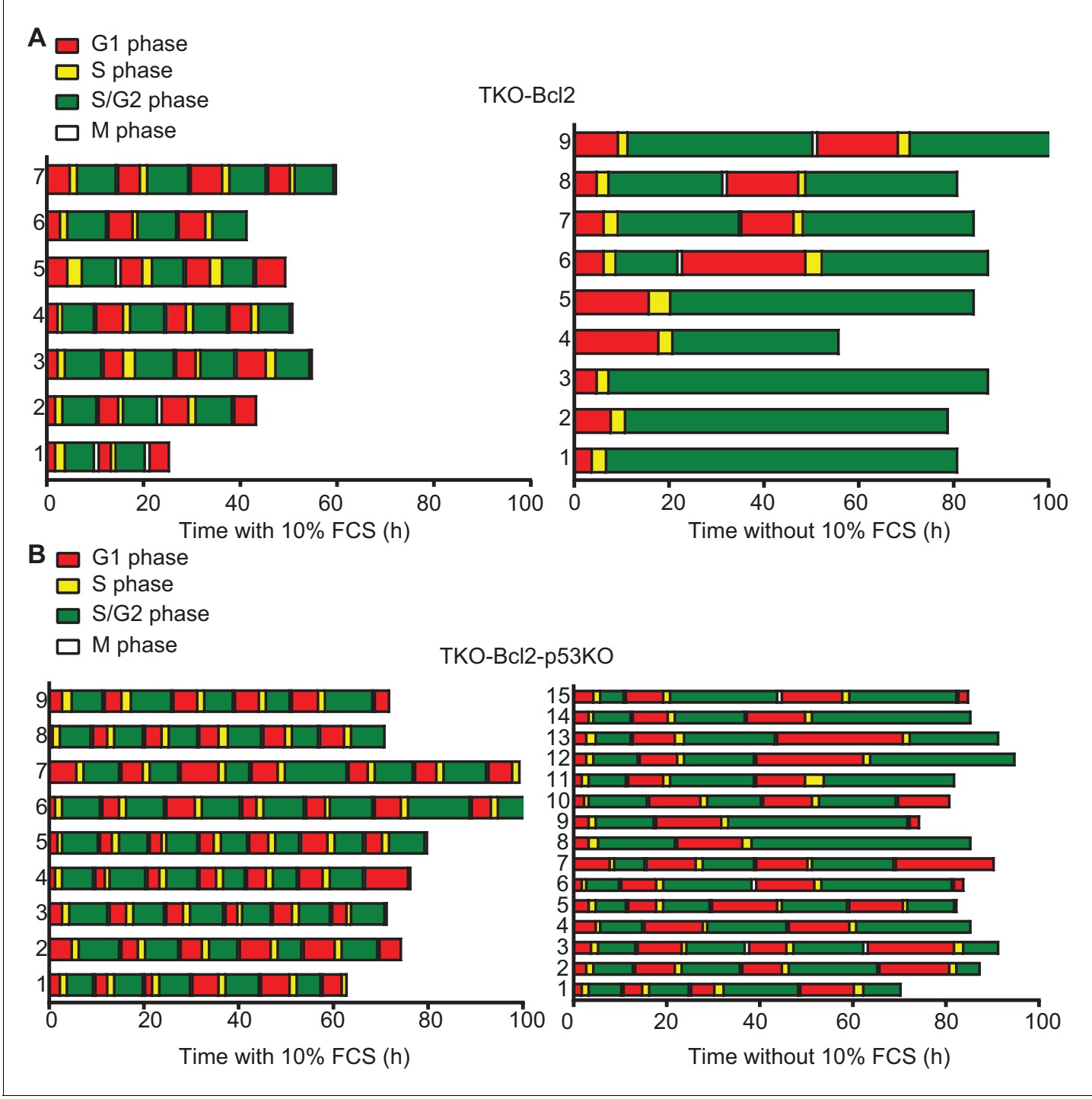

**Figure 2.** Loss of p53 rescues G2 arrest in mitogen-deprived TKO-Bcl2 MEFs. Timing of cell cycle phases in (**A**) TKO-Bcl2 MEFs and (**B**) TKO-Bcl2-p53KO MEFs expressing mKO-hCdt1 and mAG-hGem and cultured in the presence (left panels) or absence (right panels) of 10% FCS. The period a cell only expressed mKO-hCdt1 (G1 phase) is marked red, only expressed mAG-hGem (S/G2/M phase) is marked green, expressed both mKO-hCdt1 and mAG-hGem (early S phase) is marked yellow. During mitosis both markers are absent (white).The y-axes represent individual cells.

DOI: https://doi.org/10.7554/eLife.37868.005

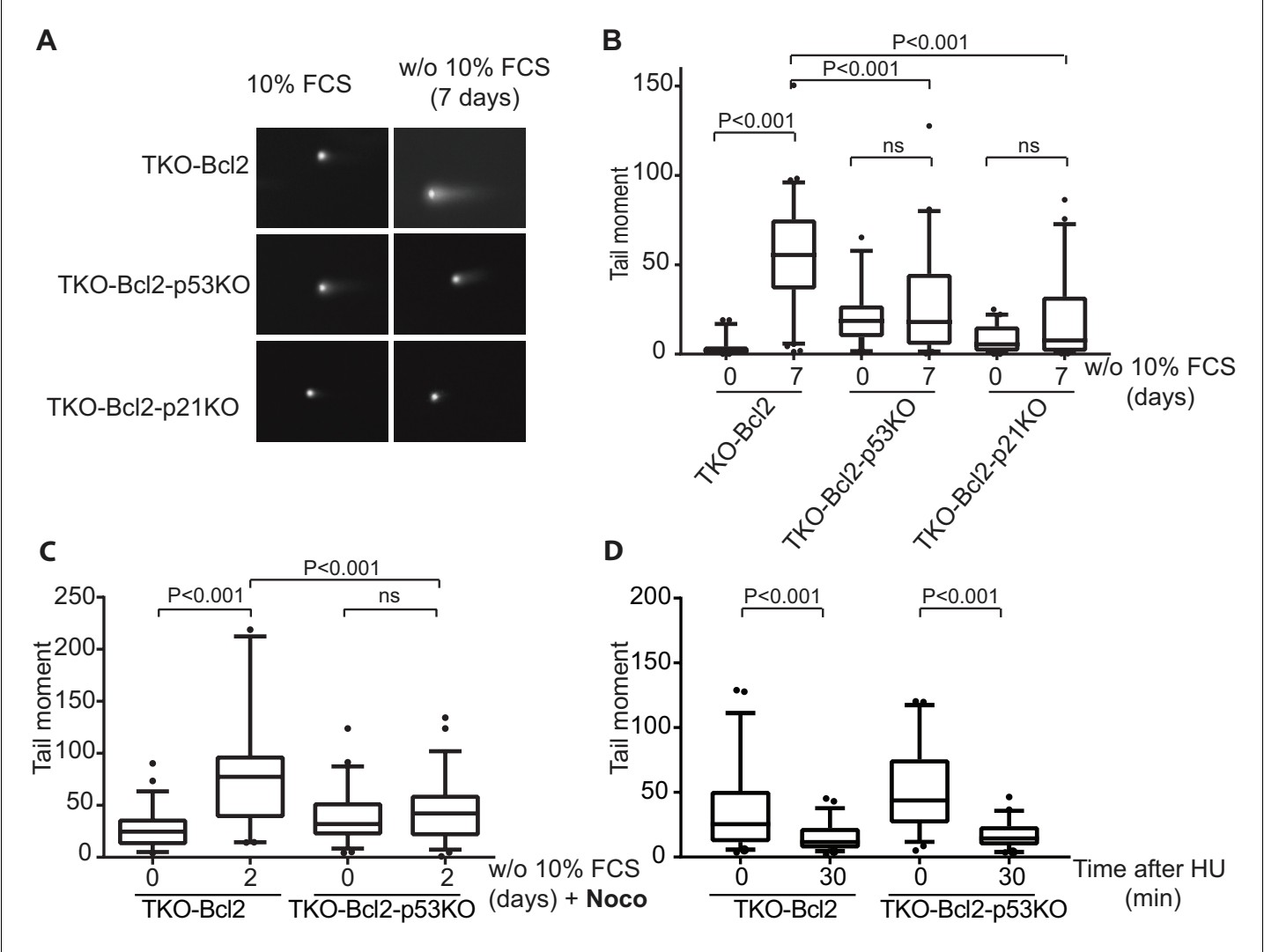

**Figure 3.** Loss of p53 reduces DNA double-stranded breaks. (**A**) Representative comets of nuclei of TKO-Bcl2, TKO-Bcl2-p53KO MEFs and TKO-Bcl2-p21KO MEFs stained with propidium iodide in the presence or absence of 10% FCS (7 days). (**B**) Tail moments obtained from TKO-Bcl2, TKO-Bcl2-p53KO and TKO-Bcl2-p21KO MEFs cultured in the presence or absence of 10% FCS (7 days). (**C**) Tail moments obtained from TKO-Bcl2 and TKO-Bcl2-p53KO MEFs cultured in the presence or absence of 10% FCS (2 days) and in the presence of nocodazole. (**D**) Tail moments obtained from TKO-Bcl2 and TKO-Bcl2-p53KO MEFs immediately and 30 min after 1 hr treatment with 2 mM HU. In B, C and D, box plots represent interquartile ranges, horizontal bars denote the median and points indicate outliers. For each condition, more than 50 cells were analyzed using the CASP software. Significance is indicated (1-way Anova nonparametric Kruskal-Wallis test).

DOI: https://doi.org/10.7554/eLife.37868.006

The following figure supplement is available for figure 3:

**Figure supplement 1.** Nocodazole-induced cell cycle arrest to prevent G1 entry.

DOI: https://doi.org/10.7554/eLife.37868.007

formation and G2-like arrest in mitogen-deprived TKO-Bcl2 MEFs. Mitogen deprivation is a strong anti-proliferative signal that may inhibit MYC transcription factors and therefore repress genes involved in nucleotide synthesis (*Gassmann et al., 1999*; *Perna et al., 2012*). Indeed, we found that mitogen-deprivation of TKO-Bcl2 cells reduced transcript levels of phosphoribosyl pyrophosphate amidotransferase (*Ppat*) and inosine monophosphate dehydrogenase 1 and 2 (*Impdh1* and *Impdh2*), genes involved in purine metabolism, 2-fold (*Figure 5—figure supplement 1A*). Reduced levels of nucleotide synthesis enzymes could impair DNA replication by disturbing the balance in the dNTP pool. Indeed, RNAi-mediated suppression of *Ppat* expression (*Figure 5—figure supplement 1B*)

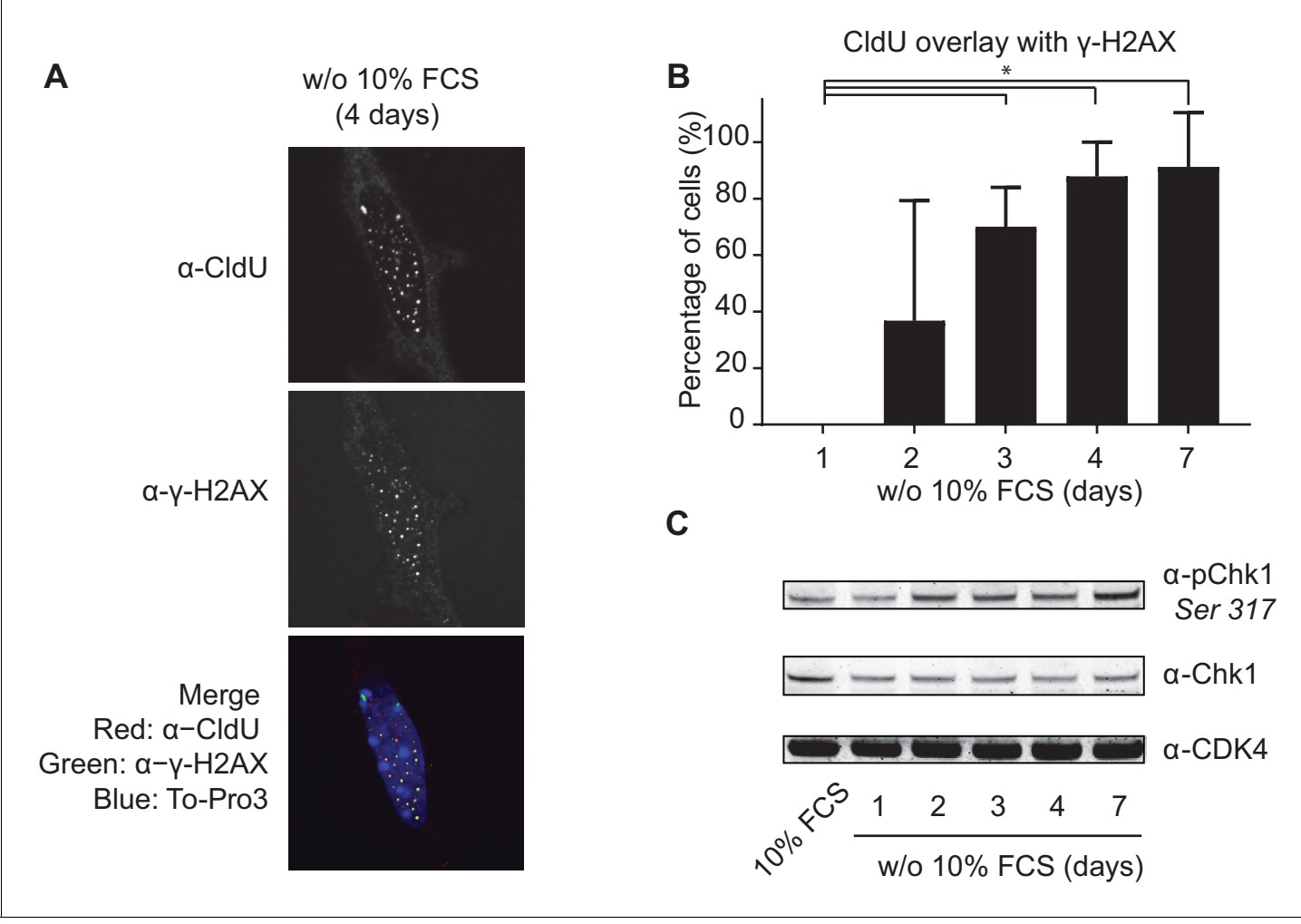

**Figure 4.** Mitogen-deprived TKO-Bcl2 MEFs suffer from replication stress. (A) CldU and γ-H2AX foci in TKO-Bcl2 MEFs cultured in the absence of 10% FCS for 4 days. DNA was labeled with To-Pro3. In the merged picture, DNA is blue, γ-H2AX is green, CldU is red and co-localization of γ-H2AX and CldU is seen as yellow foci. (B) Quantification of CldU positive TKO-Bcl2 MEFs cultured in the absence of 10% FCS for the indicated days that contained five or more superimposed γ-H2AX and CldU foci. At least 100 cells were counted per condition. Standard deviation (error bars) between at least five different microscopic slides are shown. Significant differences between average values are indicated with an asterisk (Student's t-test, p<0.01). (C) pChk1 (Chk1 phosphorylated on Ser317, upper panel) and Chk1 protein levels (middle panel) in TKO-Bcl2 MEFs cultured in the presence or absence of 10% FCS for the indicated days. Anti-CDK4 was used as loading control (lower panel).

DOI: https://doi.org/10.7554/eLife.37868.008

reduced the replication speed in mitogen-stimulated TKO-Bcl2 MEFs from 1.25 kb/min to 0.84 (shRNA #1) and 0.86 (shRNA #2) kb/min (*Figure 5—figure supplement 1C*).

Conversely, replication speed in mitogen-deprived TKO-Bcl2 MEFs could be partially rescued by the exogenous supply of nucleosides. Similar to previous experiments, one day of mitogen deprivation decreased the average fork speed by ±30%, in this experiment from 0.94 kb/min to 0.65 kb/min. In contrast, when cells were supplemented with nucleosides, reduction of replication speed was less pronounced (from 1.03 kb/min to 0.83 kb/min) (*Figure 5—figure supplement 1D*). However, daily nucleosides supplementation did not alleviate the proliferation defect of mitogen-deprived TKO-Bcl2 MEFs: G2 accumulation was hardly affected (*Figure 5—figure supplement 1E*) and also Chk1 phosphorylation and p21$^{Cip1}$ induction were not reduced (*Figure 5—figure supplement 1F*). As we observed that mitogen-independent proliferation upon *p53* loss did not require restoration of replication speed (*Figure 5B*), this indicates that rather than decelerated DNA replication another factor was causal to G2 arrest upon mitogen deprivation.

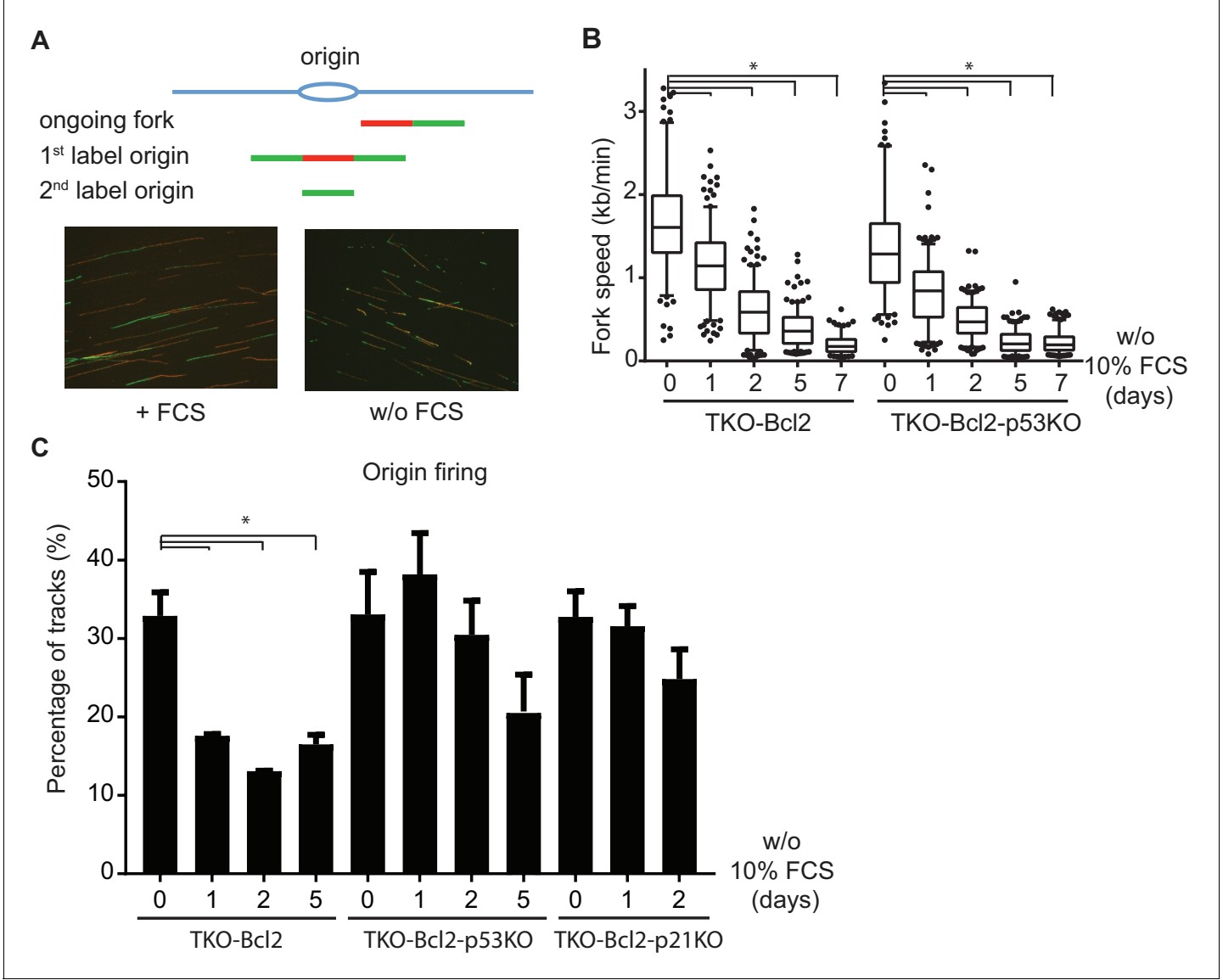

**Figure 5.** Loss of p53 restores the level of origin firing. (**A**) Schematic representation of replication tracks generated after pulse labeling with CldU (red) and IdU (green). Ongoing forks were used to determine fork speeds (kb/min); 1st label and 2nd label origins are origins of replication initiated during the labelling period with CldU and IdU, respectively (upper panel). Representative images of DNA fibers of TKO-Bcl2 MEFs with and without 10% FCS (lower panel). (**B**) Replication fork speeds in TKO-Bcl2 and TKO-Bcl2-p53KO MEFs cultured in the presence or absence of 10% FCS for 1–7 days. Box plots represent interquartile ranges, horizontal bars denote the median, whiskers indicate 5–95 percentile and points are outliers. At least 350 track lengths of ongoing forks were measured (from three independent experiments) with ImageJ. Significant differences between median values are indicated with an asterisk (nonparametric Kruskal-Wallis test, $p < 0.05$). (**C**) Quantification of origin firing in TKO-Bcl2, TKO-Bcl2-p53KO and TKO-Bcl2-p21KO MEFs cultured in the presence or absence of 10% FCS for 1–5 days. 1st label and 2nd label origins are shown as percentage of all labeled tracks (from three independent experiments). Significant differences between average values are indicated with an asterisk ($p < 0.05$, Student's t-test).

DOI: https://doi.org/10.7554/eLife.37868.009

The following figure supplements are available for figure 5:

**Figure supplement 1.** Nucleotide deficiency is not causal to G2 arrest.
DOI: https://doi.org/10.7554/eLife.37868.010

**Figure supplement 2.** DNA replication stress induced by 0.3 and 2 mM Hydroxyurea.
DOI: https://doi.org/10.7554/eLife.37868.011

## Reduced DNA breakage by p53 knockout is associated with increased origin firing

Previously, we showed that inhibition of CDK activity by p27$^{Kip1}$ and p21$^{Cip1}$ was critical for arrest of mitogen-deprived TKO-Bcl2 MEFs (*Foijer et al., 2005*). Since CDK activity is required to activate origins of replication (*Fragkos et al., 2015*; *Méndez and Stillman, 2003*), origin firing may be perturbed in mitogen-deprived TKO-Bcl2 MEFs. Indeed, among CldU/ldU-labelled DNA fibers from mitogen-deprived TKO-Bcl2 MEFs, staining patterns indicative for new origin firing were significantly reduced (*Figure 5C*). In contrast, in TKO-Bcl2-p53KO MEFs, origin firing was *not* disturbed during the first days of mitogen deprivation and maintained levels similar as in mitogen-stimulated cells (*Figure 5C*). Only after 5 days of mitogen deprivation, origin firing was reduced, which may be related to the state of confluency that was reached by that time (*Figure 1A*). Similar to TKO-Bcl2-p53KO MEFs, also TKO-Bcl2-p21KO MEFs maintained normal origin firing during the first days of mitogen deprivation (*Figure 5C*).

The increased level of origin firing upon loss of p53/p21$^{Cip1}$ contrasts to a recent publication by Roy et al. who identified a transcription-independent function of p53 in balancing replication fork homeostasis and, in contrast to our findings, observed a decrease in the level of origin firing upon loss of p53 (*Beroukhim et al., 2010*). An explanation for this seeming discrepancy may be found in comparing the different replication stress conditions. Roy et al. studied the role of p53 in conditions with a low dose of HU that did not induce DSBs. In contrast, by serum starvation we induced severe replication stress as observed by the drastic decrease of replication fork progression and induction of DNA breaks (*Figure 5B*). We therefore compared the consequences of low *versus* high doses of HU. A low dose of HU (300 µM) did not induce DNA DSBs whereas a high dose of HU (2 mM) did (*Figure 5—figure supplement 2A*). Consistent with Roy et al., loss of p53 reduced the level of origin firing upon treatment with 300 µM HU. However, loss of p53 did not change the level of origin firing after treatment with the high dose of HU (2 mM) (*Figure 5—figure supplement 2B*).

Collectively, our results suggest that under conditions of severe replication stress, restoration of the level of origin firing upon p53 loss prevents DNA breakage, allowing mitogen-independent proliferation of TKO-Bcl2-p53KO MEFs.

## Also in human cells inactivation of p53 is associated with reduced DNA DSBs

To investigate whether p53 affects DNA breakage under replication stress conditions in human cells, we used the human retinal pigment epithelial cell line RPE-1. The G1/S phase checkpoint was perturbed either by inactivating all three retinoblastoma genes, *RB*, *RBL1* and *RBL2* (TKO; *Figure 6A*) or by overexpressing a non-degradable form of human Cyclin D1 (CyclinD1; *Figure 6B*). Overexpression of Cyclin D1 is biologically relevant since the gene encoding Cyclin D1 represents the second most frequently amplified locus in the human cancer genome (*Beroukhim et al., 2010*). In addition, in many human tumors overexpression of D type cyclins takes place in the absence of detectable genomic alterations (*Hosokawa and Arnold, 1998*). In the presence of mitogens, TKO and CyclinD1 RPE-1 cells proliferated faster than wild type RPE-1s (*Figure 6—figure supplement 1A*). 24 hr after mitogen-deprivation, wild type RPE-1s arrested in the G1 phase of the cell cycle (*Figure 6C*), whereas both, TKO and CyclinD1 cultures maintained a normal cell cycle profile up to 72 hr (*Figure 6D and E*, respectively). Upon prolonged mitogen starvation for more than 4 days, TKO and CyclinD1 cells started to die (*Figure 6—figure supplement 1B*). Cell death could not be avoided by overexpression of Bcl2 (*Figure 6—figure supplement 1C, D and E*) nor by additional inactivation of *TP53* (*Figure 6F,G* and *Figure 6—figure supplement 1F*). Apparently, RPE-1 cells lacking the G1/S phase checkpoint were very sensitive to apoptosis in the absence of mitogenic stimulation, which could not easily be suppressed. Nonetheless, we could follow the behavior of these cells during the first days of mitogen deprivation. Similar to TKO-Bcl2 MEFs, p53-proficient TKO-Bcl2 RPE-1s showed induction of DNA DSBs after one day of mitogen starvation. In contrast, no DSB induction was seen in TKO-Bcl2-p53KO RPE-1s (*Figure 6H*). Similarly, mitogen starvation hardly induced DSBs in CyclinD1-Bcl2-p53KO RPE-1s compared to CyclinD1-Bcl2 RPE-1s (*Figure 6I*). Mitogen-deprived TKO-Bcl2 and CyclinD1-Bcl2 RPE-1s showed a decrease in the level origin firing (*Figure 6J,K*). In contrast, TKO-Bcl2-p53KO and CyclinD1-Bcl2-p53KO RPE1-s maintained normal levels of origin

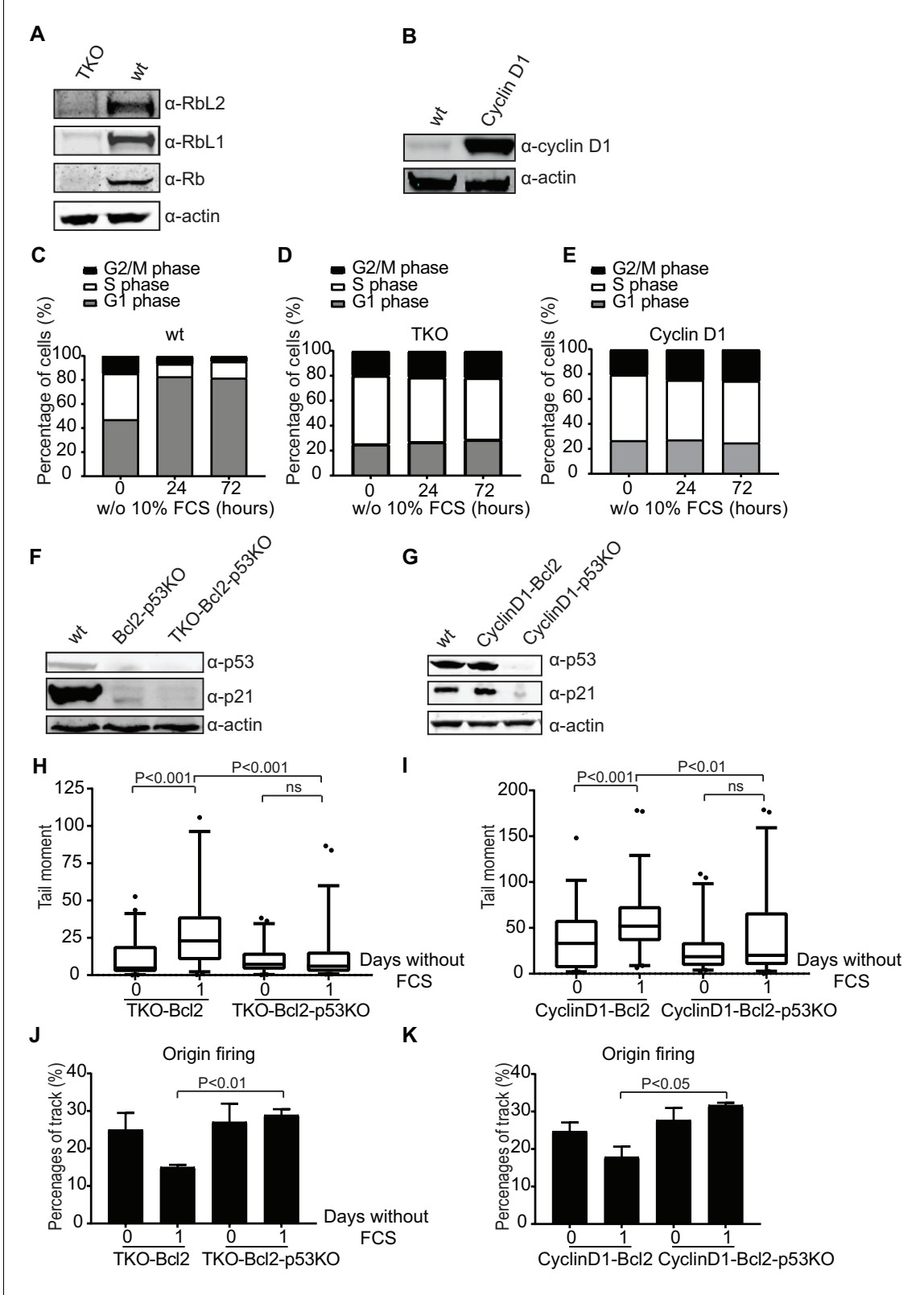

**Figure 6.** Loss of p53 reduces DNA double-strand breaks in human cells. (**A**) Rb, Rbl1 and Rbl2 protein levels in wt and TKO RPE-1s. Anti-actin was used as a loading control. (**B**) Cyclin D1 protein levels in wt and CyclinD1 RPE-1s. Anti-actin was used as a loading control. (**C, D, E**) Cell cycle distribution based on propidium iodide content of wt (**C**), TKO (**D**) and CyclinD1 (**E**) RPE-1s in the absence of 10% FCS for the indicated hours. (**F**) p53 and p21$^{Cip1}$ protein levels in wt, Bcl2-p53KO and TKO-Bcl2-p53KO RPE-1s. Anti-actin was used as a loading control. (**G**) p53 and p21$^{Cip1}$ protein levels

*Figure 6 continued on next page*

*Figure 6 continued*

in wt, CyclinD1-Bcl2 and CyclinD1-Bcl2-p53KO RPE-1s. Anti-actin was used as a loading control. (H, I) Tail moments obtained from TKO-Bcl2, TKO-Bcl2-p53KO (H) and CyclinD1-Bcl2 and CyclinD1-Bcl2-p53KO (I) RPE1-s cultured in the presence or absence of 10% FCS (1 day). Box plots represent interquartile ranges, horizontal bars denote the median and points are outliers. For each condition, more than 50 cells were analyzed using the CASP software. Significance is indicated (1-way Anova nonparametric Kruskal-Wallis test). (J, K) Quantification of origin firing in TKO-Bcl2, TKO-Bcl2-p53KO (J) and CyclinD1-Bcl2 and CyclinD1-Bcl2-p53KO (K) RPE1-s cultured in the presence or absence of 10% FCS for 1 day. 1st label and 2nd label origins are shown as percentage of all labelled tracks (from two independent experiments). Significant differences between average values are indicated (Student's t-test).

DOI: https://doi.org/10.7554/eLife.37868.012

The following figure supplement is available for figure 6:

**Figure supplement 1.** Response of RPE1 cells to mitogen starvation

DOI: https://doi.org/10.7554/eLife.37868.013

firing after mitogen deprivation (*Figure 6J,K*), although there is no difference in replication fork speed upon mitogen-deprivation (*Figure 6—figure supplement 1G and F*).

These results show that also in human cells inactivation of p53 in G1/S phase checkpoint defective cells reduced the accumulation of DNA DSBs following mitogen deprivation, possible by rescuing the level of origin firing.

## The role of *p53* loss in tumor development

To investigate whether the accumulation of DNA DSBs also operates *in vivo* to impede tumor growth, we studied retinoblastoma development in chimeric mice generated by blastocyst injection of $Rb^{-/-}Rbl2^{-/-}$ embryonic stem cells (ESCs) (*Dannenberg et al., 2000*). Remarkably, murine retinoblastomas showed pronounced p53 and γ-H2AX staining (*Figure 7A*). However, by sequencing, *p53* appeared wild-type in a separate series of seven tumors, indicating that in this model retinoblastomas did activate the DDR but could still colonize the entire eyeball. To study if *p53* inactivation accelerates tumorigenesis, we inactivated *p53* in $Rb^{-/-}$ $Rbl2^{-/-}$ ESCs using CRISPR/Cas9-mediated gene disruption. However, no chimeric animals were obtained from $Rb^{-/-}Rbl2^{-/-}p53^{-/-}$ ESCs, likely indicating that combined ablation of the Rb and p53 pathways is incompatible with embryonic development.

As an alternative *in vivo* readout, we injected $Rb^{-/-}Rbl2^{-/-}$ and $Rb^{-/-}Rbl2^{-/-}p53^{-/-}$ mouse ESCs under the skin of nude mice. $Rb^{-/-}Rbl2^{-/-}$ ESCs developed a teratoma in 4 out of 6 mice; in contrast, $Rb^{-/-}Rbl2^{-/-}p53^{-/-}$ ESCs developed a tumor in 6 out of 6 injected mice. On average the $Rb^{-/-}Rbl2^{-/-}p53^{-/-}$ tumors were larger than $Rb^{-/-}Rbl2^{-/-}$ tumors (*Figure 7B*), although there is a 11% chance that the difference is accidental (p=0.1116, unpaired t-test). Teratomas of both genotypes mainly showed early neuronal differentiation and stained positive for the replication stress marker γ-H2AX (*Figure 7C*), suggesting that all tumors were suffering from replication stress. To assess the presence of DSBs, we performed a neutral comet assay on teratoma tissues. Three of the four $Rb^{-/-}Rbl2^{-/-}$ teratomas showed an increase in tail moment compared to the tail moments of $Rb^{-/-}Rbl2^{-/-}p53^{-/-}$ teratomas (*Figure 7D*). Of note, unlike the other tumors, the largest $Rb^{-/-}Rbl2^{-/-}$ tumor (marked with asterisk in *Figure 7B,D*) had high levels of infiltrating neutrophils, which possibly explains its bigger size as well as the low level of DNA DSBs.

Although the number of tumors was small, p53 knockout teratomas showed a trend towards lower levels of DSBs and accelerated tumor growth. Therefore, both our *in vitro* as well as *in vivo* data suggest that inactivation of p53 in G1/S checkpoint deficient cells contributes to tumorigenesis by reducing DNA DSBs (*Figure 7E*).

## Discussion

We have previously shown that apoptosis-resistant MEFs that lack the G1/S phase checkpoint (TKO-Bcl2 MEFs) can undergo unscheduled S-phase entry. Here we show that they do so at the expense of severe replication stress and the accumulation of DNA DSBs, which ultimately causes G2-like cell cycle arrest. Inactivation of *p53* allowed mitogen-independent proliferation, which remarkably was not only associated with alleviated G2 arrest but also with reduced DNA breakage and restored origin firing. The firing of origins requires Cyclin-CDK activity (*Fragkos et al., 2015*; *Méndez and*

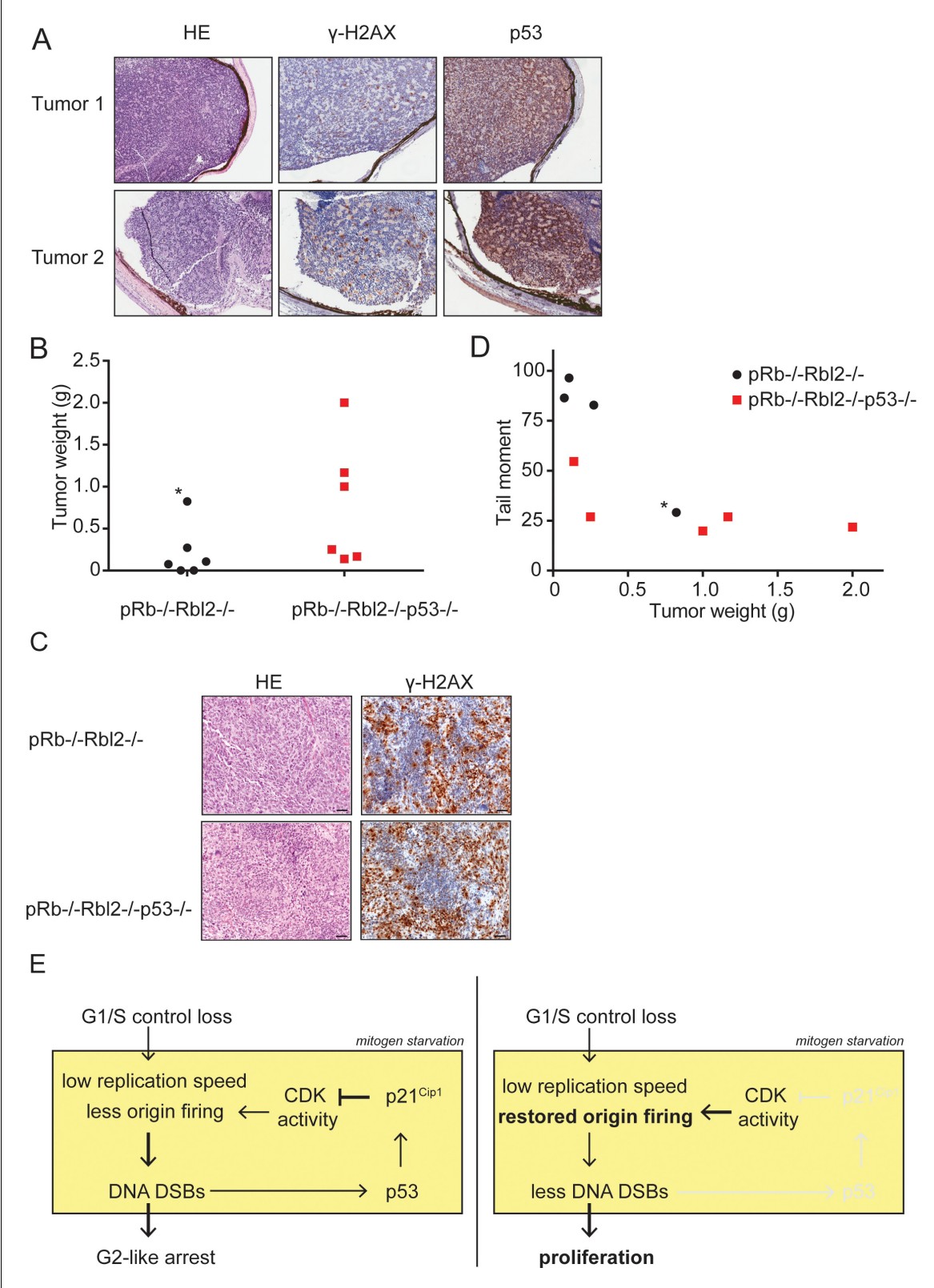

**Figure 7.** p53 inactivation promotes tumor growth *in vivo*. (**A**) Examples of HE and immunohistochemical stainings for γ-H2AX and p53 of two retinoblastomas from $Rb^{-/-}Rbl2^{-/-}$ chimeric mice. (**B**) Tumor weight of $Rb^{-/-}Rbl2^{-/-}$ (black) and $Rb^{-/-}Rbl2^{-/-}p53^{-/-}$ (red) teratomas 20 days after injection of ESCs in nude mice. Black dot marked with an asterisk (*) indicates the only tumor with high levels of infiltrating neutrophils. (**C**) Tail moments of $Rb^{-/-}Rbl2^{-/-}$ (black) and $Rb^{-/-}Rbl2^{-/-}p53^{-/-}$ (red) teratomas plotted against the tumor weight. Spearman's correlation coefficient between tail moment and

*Figure 7 continued on next page*

Figure 7 continued

tumor weight is −0.78. Black dot marked with an asterisk (*) indicates the only tumor with high levels of infiltrating neutrophils. One $Rb^{-/-}Rbl2^{-/-}p53^{-/-}$ (red) teratoma could not be analyzed for DNA DSBs due to the small tissue size. For each teratoma, more than 50 cells were analyzed using the CASP software. (D) Examples of HE and immunohistochemical staining for γ-H2AX of $Rb^{-/-}Rbl2^{-/-}$ and $Rb^{-/-}Rbl2^{-/-}p53^{-/-}$ teratomas. Scale bar is 50 μm. (E) Schematic model for how p53 inactivation reduces DNA DSBs in mitogen-deprived cells lacking the G1/S checkpoint. Cells that lost the G1/S phase checkpoint suffer from replication stress leading to DNA DSBs. Activation of p53 and p21$^{Cip1}$ inhibits CDK activity and thereby inhibits the firing of new origins leading to more DNA damage and establishment of the G2-like arrest (left panel). Inactivation of p53 and therefore its downstream protein p21$^{Cip1}$ increases CDK activity and allows origins to fire. Restored levels of origin firing rescues stalled forks, causing less DNA DSB formation and enabling proliferation (right panel).

DOI: https://doi.org/10.7554/eLife.37868.014

Stillman, 2003). In mitogen-deprived TKO-Bcl2 MEFs, CDK activity was low due to the high levels of p21$^{Cip1}$ and p27$^{Kip1}$ explaining the low level of origin firing. It is therefore likely that upon p53 inactivation and therefore reduction of p21$^{Cip1}$, CDK activity increased (Foijer et al., 2005) and hence promoted origin firing. Consistently, we found that also genetic inactivation of p21$^{Cip1}$ restored origin firing and promoted proliferation of mitogen-deprived TKO-Bcl2 MEFs. Importantly, this phenomenon was not restricted to murine cells, but also observed in human RPE-1 cells: mitogen deprivation restricted origin firing and induced DNA breakage in G1/S-checkpoint-defective RPE-1s, and both could be reverted by inactivation of p53. However, for as yet unknown reasons, RPE-1 cells appeared highly sensitive to apoptosis and therefore the damage-reducing effect of p53 loss did not translate into mitogen-independent proliferation.

While restored origin firing upon ablation of the p53/p21$^{Cip1}$ axis is mechanistically plausible, we have not directly proven that DNA breakage as a consequence of replication stress was prevented by increased origin firing. Related to this, an important question is whether p53 inactivation reduced the formation of DNA breaks or stimulated DSB repair. Apart from its role as transcription factor, p53 has many transcription-independent functions, among which inhibition of DNA DSB repair by both non-homologous end joining (NHEJ) and homologous recombination (HR) (Menon and Povirk, 2014; Sengupta and Harris, 2005; Akyüz et al., 2002; Dudenhöffer et al., 1998). However, we show that KO of the p53 transcription target p21$^{Cip1}$ phenocopied the effects of KO of p53 in TKO-Bcl2 MEFs, arguing against a transcription-independent role of p53 in suppressing DNA repair. Furthermore, increased DNA repair by NHEJ in G1 phase is unlikely as the levels of DSBs in serum-starved TKO-Bcl2-p53KO MEFs remained low when G1 entry was prevented by artificially arresting cells in M-phase. In this experiment it remains possible that an increase in HR activity during S/G2 phase contributed to less DNA breaks in mitogen-deprived p53KO MEFs. Also this possibility seems unlikely as the repair of HU-induced DSBs was not affected by p53 status. However, this experiment does not exclude the possibility that under mitogen-deprived conditions p53 suppressed DSB repair. With this restriction, we hypothesize that abrogation of p53 reduced the formation of DNA breaks, rather than facilitated repair.

Novel roles of pRB and p53 are emerging but it is unclear to which extent they are implicated in suppression of cancer. Apart from its well documented role in cell cycle control, pRB has emerged as a multi-functional protein involved in a wide range of biological processes including chromatin architecture, cohesion, chromosome condensation during mitosis, DNA replication via interaction with replication components and involvement in DNA repair processes such as HR and NHEJ (Vélez-Cruz and Johnson, 2017; Dick and Rubin, 2013; Huang et al., 2015). We suggest that these other functions of pRB do not play a role in the accumulation of DNA damage in Rb-deficient cells since we observed the same phenotype in Rb-proficient Cyclin D1 overexpressing RPE-1s. Thus, the accumulation of DNA DSBs in mitogen-deprived conditions can be attributed to loss of the G1/S phase checkpoint and not to other functions of the pRB protein or its family members.

It has been described that some ribosomal proteins have a function in the DNA damage response that is activated upon intrinsic replication stress and mediated through the Mdm2-p53 axis (Xu et al., 2016). In addition, some ribosomal proteins act as a sensor for DNA damage and directly participate in the process of DNA repair. In this study, we cannot exclude an effect of p53 loss on the extra-ribosomal functions of these proteins. Also p53 has functions outside its canonical role in the DDR. Recently, a novel transcription-independent role for p53 in balancing replication homeostasis was reported. The p53 protein can bind to replication forks and facilitate replication fork restart

in replication stress conditions (*Roy et al., 2018*). Since we found a transcription-dependent role of p53 in suppressing origin firing, we hypothesize that the two different effects of p53 loss reflect different functions of p53 that operate side by side: dependent on the severity of replication stress, p53 facilitates replication fork restart and suppresses the firing of new origins.

Others have shown that disruption of the nucleotide pool can contribute to replication stress, DNA breakage and cell death in oncogene-expressing cells (*Bester et al., 2011*; *Poli et al., 2012*; *Beck et al., 2012*; *Pfister et al., 2015*). We were able to partially rescue replication speed in mitogen-deprived TKO-Bcl2 MEFs by exogenous supply of nucleosides. However, increased replication speed was not sufficient to overcome G2 arrest and to support normal cell cycle progression. Furthermore, we found that despite the capacity of TKO-Bcl2-p53KO MEFs to proliferate mitogen-independently, replication speed was still reduced. These observations indicate that another factor rather than the speed of DNA synthesis was critical for DNA breakage and G2 arrest in mitogen-deprived TKO-Bcl2 cells. As only origin firing but not replication speed was affected by *p53* status, we favor a scenario where restoration of origin firing upon inactivation of *p53*, given the involvement of p21$^{Cip1}$ likely as a result of restored CDK activity, suppressed DNA breakage and allowed mitogen-independent proliferation.

Loss of the Rb and p53 pathways frequently occur and co-occur in human tumors (*Polager and Ginsberg, 2009*). The p53 gene is mutated in more than 50% of human cancer, and mutations in other genes that affect p53 function occur in many, if not all, tumors that retain a normal p53 gene (*Perri et al., 2016*). In addition, most human tumors lack the G1/S phase checkpoint. For example many human tumors overexpress D-type cyclins and *CCND1* represents the second most frequently amplified locus in the human cancer genome (*Hosokawa and Arnold, 1998*; *Menon and Povirk, 2014*). Furthermore, evading apoptosis is one of the hallmarks of cancer and the anti-apoptotic gene Bcl2, which is used in this study to suppress apoptosis, is commonly overexpressed in many types of cancer, including renal, prostate, gastric, lung and colorectal cancer, neuroblastoma, non-Hodgkin's lymphoma and acute and chronic leukemia (*Frenzel et al., 2009*; *Kirkin et al., 2004*). Thus, most tumor cells harbor the type of mutations used in this study. Whereas we can only speculate about the precise number of cancer types that harbor the exact combination of Rb, p53 and Bcl2 aberrations as used in this study, there are examples known. For example, approximately 90% of small cell lung cancer tumors have lost both p53 and Rb (*Sekido et al., 2003*). Beside this, small cell lung tumors are also characterized by expression of Bcl2 (*Kaiser et al., 1996*). Furthermore, human retinoblastoma originates from an intrinsic death-resistant precursor cell (*Xu et al., 2009*), is characterized by mutations in the Rb gene and it is suggested that the p53 pathway is inactivated (*Xu et al., 2009*; *Laurie et al., 2006*). Although *p53* mutations are infrequent in human retinoblastomas, the p53 pathway may be intrinsically attenuated upon *RB1* loss by miR-24-mediated downregulation of p14$^{ARF}$ (*To et al., 2012*) and by NANOS-mediated suppression of p53-activating kinases (*Miles et al., 2014*). Other studies suggested that *RB1*-deficient retinal cells achieve attenuation of the p53 pathway by high expression of MDM2 and MDMX (*Xu et al., 2009*; *Laurie et al., 2006*), although a recent paper revealed critical p53-independent functions of high MDM2 expression (*Qi and Cobrinik, 2017*).

In our chimeric mouse model of retinoblastoma, we found evidence for DNA damage, but loss of *p53* was not a requirement for development of eye-filling tumors. Unfortunately, we could not study the effect of *p53* loss, but using a hereditary retinoblastoma model, others reported a dramatic effect of *p53* inactivation. When *Rb* was conditionally inactivated in retinal progenitor cells in a *p107*$^{-/-}$ background, non-invasive retinoblastomas developed with a penetrance of 60%. Upon additional inactivation of *p53*, aggressive, invasive bilateral retinoblastomas developed with 100% penetrance and reduced latency (*Dyer et al., 2005*; *Zhang et al., 2004*). Importantly, evidence has been obtained that murine retinoblastomas originate from an intrinsically death-resistant cell of origin (*Chen et al., 2004*). We therefore propose that the tumor promoting effect of attenuated p53 activity was not due to abrogation of an apoptotic response but rather required for maintaining sufficient CDK activity to counteract the deleterious effects of replication stress. We obtained support for such tumor-promoting effect of *p53* ablation in an *Rb*$^{-/-}$*Rbl2*$^{-/-}$ teratoma model: tumor size was inversely correlated with the level of DNA breaks and *Rb*$^{-/-}$*Rbl2*$^{-/-}$*p53*$^{-/-}$ teratomas generally showed lower levels of DNA breaks than *Rb*$^{-/-}$*Rbl2*$^{-/-}$ teratomas.

Finally, our results are likely related to intriguing observations that at least for some tumor types the outgrowth of early cancerous lesions is prohibited by activation of the DDR (*Bartkova et al.,*

*2006*). It has been suggested that oncogene activation can directly cause replication stress by hyper-stimulating DNA replication, which activates the ATR-Chk1 axis (*Hills and Diffley, 2014*; *Di Micco et al., 2006*; *Halazonetis et al., 2008*). Furthermore, frequent DNA breakage associated with replication stress activates the complementary ATM-Chk2-p53 module that provides a strong barrier to proliferation by inducing apoptosis or permanent cell cycle arrest. It has therefore been suggested that activation of the DDR may explain the strong selective pressure for loss of p53 in human cancer (*Halazonetis et al., 2008*). Rather than a direct consequence of oncogene activation, replication stress in our system was the consequence of the combination of Rb-protein deficiency (hyper-activating E2F transcription factors) and growth-restricting conditions (the absence of mitogenic signaling), leading to DNA breakage and activation of the DDR. Furthermore, we found loss of p53 not only abrogated cell cycle arrest and apoptosis, but also suppressed the induction of DNA damage itself, providing a novel mechanistic explanation for the frequent co-occurrence of p53 and pRb pathway inactivation in cancer.

## Materials and methods

**Key resources table**

| Reagent type (species) or resource | Designation | Source or reference | Identifiers | Additional information |
|---|---|---|---|---|
| Cell line (*mus musculus*) male | TKO-Bcl2 mouse embryonic fibroblasts (MEFs) | PMID:16338659 | | |
| Cell line (*mus musculus*) male | TKO-Bcl2-p53KO MEFs | This paper | N/A | Knockout of p53 in TKO-Bcl2 MEFs, described in the materials and methods |
| Cell line (*mus musculus*) male | TKO-p53RNAi MEFs | PMID:16338659 | | |
| Cell line (*mus musculus*) male | TKO-p53KO MEFs | This paper | N/A | Knockout of p53 in TKO MEFs, described in the materials and methods |
| cell line (*mus musculus*) male | $Rb^{-/-}Rbl2^{-/-}$ mESCs | PMID: 15574596 | | |
| Cell line (*mus musculus*) male | $Rb^{-/-}Rbl2^{-/-}p53^{-/-}$ mESCs | This paper | | Knockout of p53 in $Rb^{-/-}Rbl2^{-/-}$, described in the materials and methods |
| Cell line (*homo sapiens*) female | RPE-1 hTERT | ATCC | Cat# CRL-4000 | RRID:CVCL_4388 |
| Cell line (*homo sapiens*) female | TKO RPE-1s | This paper | N/A | Knockout of Rb, Rbl1 and Rbl2 in RPE-1-hTERT cells, described in material and methods |
| Cell line (*homo sapiens*) female | TKO-Bcl2 RPE-1s | This paper | N/A | Knockout of Rb, Rbl1, Rbl2 and overexpression of Bcl2 in RPE-1-hTERT cells, described in material and methods |
| Cell line (*homo sapiens*) female | TKO-Bcl2-p53KO RPE-1s | This paper | N/A | Knockout of Rb, Rbl1, Rbl2 and TP53 and overexpression of Bcl2 in RPE-1-hTERT cells, described in material and methods |

*Continued on next page*

*Continued*

| Reagent type (species) or resource | Designation | Source or reference | Identifiers | Additional information |
|---|---|---|---|---|
| Cell line (*homo sapiens*) female | CyclinD1 RPE-1s | This paper | N/A | Overexpression of non-degradable CyclinD1 (T286A) in RPE-1-hTERT cells, described in material and methods |
| Cell line (*homo sapiens*) female | CyclinD1-Bcl2 RPE-1s | This paper | N/A | Overexpression of non-degradable CyclinD1 (T286A) and Bcl2 in RPE-1-hTERT cells, described in material and methods |
| Cell line (*homo sapiens*) female | CyclinD1-Bcl2-p53KO RPE-1s | This paper | N/A | Knockout of TP53 and overexpression of non-degradable CyclinD1 (T286A) and Bcl2 in RPE-1-hTERT cells, described in material and methods |
| Antibody | Brdu (recognizing CldU) | Bioconnect | Cat# OBT0030G Clone BU1/75 | RRID: AB_609567 (1/500) |
| Antibody | BrdU (recognizing IdU) | Becton Dickinson | Cat# 347580 Clone B44 | RRID: AB_10015219 (1/750) |
| Antibody | mouse p53 | Abnova Corporation | Cat# MAB9657 Clone IMX25 | RRID: AB_10756365 (1/1000) |
| Antibody | mouse/ human p21 | Santa Cruz | Cat# sc-397 C19 | RRID: AB_632126 (1/500) |
| Antibody | p27 | BD Transduction Laboratory | Cat# 554069 | RRID: AB_395225 (1/2000) |
| Antibody | CDK4 | Santa Cruz | Cat# sc-260 C22 | RRID: AB_631219 (1/2000) |
| Antibody | Actin | Santa Cruz | Cat# sc-1616 | RRID:AB_630836 (1/1000) |
| Antibody | pChk1 Ser 317 | Bethyl | Cat# A304-673A | RRID: AB_2620868 (1/1000) |
| Antibody | Chk1 | Santa Cruz | Cat# sc-8408 G4 | RRID: AB_627257 (1/1000) |
| Antibody | Rb1 | Santa Cruz | Cat# sc-50 C15 | RRID: AB_632339 (1/500) |
| Antibody | Rbl1 | Santa Cruz | Cat# sc-318 C18 | RRID: AB_2175428 (1/1000) |
| Antibody | Rbl2 | Lab Vision | Cat# OP117 Clone AB1 | RRID: AB_145257 (1/1000) |
| Antibody | Cyclin D1 | Santa Cruz | Cat# sc-753 H296 | RRID: AB_2070433 (1/1000) |
| Antibody | human p53 | BD Bioscience | Cat# 554293 Clone DO-1 | RRID: AB_395348 (1/500) |
| Antibody | Bcl2 | Santa Cruz | Cat# sc-509 | RRID: AB_626733 (1/1000) |
| Chemical compound, drug | 2'-deoxyadenosine monohydrate (dA) | Jena Bioscience | N-DN-1001 | |
| Chemical compound, drug | 2'-deoxycytidine monohydrochloride (dC) | Jena Bioscience | N-DN-6352 | |
| Chemical compound, drug | 2'-deoxyguanosine monohydrate (dG) | Jena Bioscience | N-DN-1003 | |
| Chemical compound, drug | 2'-deoxythymidine (dT) | Jena Bioscience | N-DN-6354 | |

*Continued on next page*

*Continued*

| Reagent type (species) or resource | Designation | Source or reference | Identifiers | Additional information |
|---|---|---|---|---|
| Chemical compound, drug | Cell Player 96-well kinetic caspase-3/7 reagent | Essen Bioscience | Cat# 4440 | |
| Chemical compound, drug | CldU | Sigma | Cat# C6891-100mg | |
| Chemical compound, drug | IdU | Sigma | Cat# I7125-5g | |
| Chemical compound, drug | BrdU | Sigma | Cat# B5002-1G | |
| Chemical compound, drug | Hydroxyurea (HU) | Sigma | Cat# H8627 | |
| Chemical compound, drug | Propidium Iodide (PI) | Thermo fisher scientific | Cat# P3566 | |
| Chemical compound, drug | Puromycin | Sigma | Cat# P7255 | |
| Chemical compound, drug | Penincillin/ Streptomycin | Gibco/Life technologies | Cat# 15140122 | |
| Chemical compound, drug | Lipofectamine 2000 | Invitrogen | Cat# 11668030 | |
| Chemical compound, drug | Polybrene | Sigma | Cat# H9268 | |
| Chemical compound, drug | Fluor-gel with TES buffer | Electron Microscopy Science | Cat# 17985–30 | |
| Software, algorithm | GraphPad Prism 7 | GraphPad Software | https://www.graphpad.com/scientific-software/prism/ | |
| Software, algorithm | Adobe Photoshop CS6 | Adobe | https://www.adobe.com/products/photoshop.html | |
| Software, algorithm | Adobe Illustrator CS6 | Adobe | https://www.adobe.com/products/illustrator.html | |
| Software, algorithm | Image Studio Lite Ver. 4.0 | LI-COR Biosciences | https://www.licor.com/bio/products/software/image_studio_lite/ | |
| Software, algorithm | ImageJ | | https://imagej.nih.gov/ij/download.html | |
| Software, algorithm | FlowJo software version 7.6.1 | FlowJo, LCC | https://www.flowjo.com/solutions/flowjo/downloads | |
| Software, algorithm | Casplab | Casplab | http://casplab.com/download | |

## Cell culture

MEFs were isolated from chimeric embryos as previously described (*Dannenberg et al., 2000*) and cultured in GMEM (Invitrogen), supplemented with 10% fetal calf serum (FCS), 0.1 mM nonessential amino acids (Invitrogen), 1 mM sodium pyruvate (Invitrogen), 100 μg/ml penicillin, 100 μg/ml streptomycin (Invitrogen) and 0.1 mM β-mercaptoethanol (Merck) in the absence or presence of nucleoside (200 nM of Cytidine, Guanosine, Adenosine and Thymidine). TKO-Bcl2 overexpressing MEFs

and TKO-p53RNAi were generated as described previously (*Foijer et al., 2005*). CRISPR/Cas9 technology was used to inactive *Trp53* and *Cdkn1a*.

RPE-1 cells were kindly provided by J. Raaijmakers, who purchased the cells from ATCC. RPE-1 cells were cultured in DMEM/F12+GlutaMAX (Invitrogen), supplemented with 10% FCS, 100 µg/ml penicillin and 100 µg/ml streptomycin (Invitrogen). CRISPR/Cas9 technology was used to inactivate *Rb*, *Rbl1*, *Rbl2 and TP53*. Bcl2 cDNA and a non-degradable form of CylcinD1 cDNA was overexpressed using retroviral transfection.

For serum starvation experiments, cells were trypsinized and allowed to attach in the presence of serum for 4 hr. Subsequently, cells were washed with PBS and supplemented with serum free medium. To block progression into mitosis, cells were cultured in the presence of 250 ng/ml nocodazole. All cell lines have been tested for mycoplasma (PCR).

## Constructs, transfections, lentiviral and retroviral infections

The FUCCI constructs CSII-EF-MCS-mKO-hCdt1 (30/120) and CSII-EF-MCS-mAG-hGem (1/110) were kindly provided by A. Miyawaki (*Sakaue-Sawano et al., 2008*). The 19-mer *Trp53* targeting sequence in pRetroSuper-RNAi-p53 is GTACATGTGTAATAGCTCC (*Foijer et al., 2005*). Gene specific guideRNAs (mouse *Trp53*: TACCTCTCTTTGCGCTCCCT (*Platt et al., 2014*); human *RB* TGAAC-GACATCTCATCT, human *RBL1* TTTCGTGAACGTATAGAA, human *RBL2* CGAGGTTGCTCCTC TTGA and human *TP53* GACGCTAGGATCTGACTG) were annealed to generate short double-strand DNA fragments with four base pairs overhang (CACC and AAAC) compatible with ligation into the *BbsI* digested Cas9/CRISPR px330-puro plasmid. The px330-p53 guideRNA vector was transfected into MEFs using Polyethylenimine (PEI) transfections. The px330-Rb-, px330-Rbl1-, px330-Rbl2 and px330-p53 guideRNA vectors were transfected into RPE-1 cells using Lipofectamine 2000 (Invitrogen). Afterwards, RPE-1 cells were selected with 10 ug/ml puromycin for two days. Two specific guideRNAs targeting the mouse *Cdkn1a* gene were AGCGCAGATTGGTCTTCT and CCCGCAGCCGTGACGACC with four base pairs overhang (CACC and AAAC) compatible with ligation into the in *BmsbI* digested pLentiCRISPR v2 vector. The 21-mer oligos in pLKO.1 targeting *Ppat* were: #1: CCACATGCTTATGTATGTATA and #2: CCGGAGAAATTGTAGAAATAT. Corresponding empty vector (EV) was used as control. Lentiviral plasmids were co-transfected with the helper plasmids pMDLgpRRE, VSV-G and pRSV-Rec into HEK293T cells by PEI transfection. A pBABE-puro retroviral vector encoding a non-degradable form of Cyclin D1 (T286A) was kindly provided by R. Agami (*Agami and Bernards, 2000*). This retroviral vector was co-transfected with the helper plasmids puMCV-Gag pol MMLV and pCMV VSVG into HEK293T cells by PEI transfection. Both for lentiviral and retroviral transfections, forty-eight and sixty-two hours post transfection viral supernatants were filtered through 0.45 µm filter and used to infect MEFs in the presence of 4 µg/ml polybrene three times for 8–12 hr.

## Growth curves and caspase assay

The IncuCyte ZOOM instrument (Essen Bioscience) live cell imaging system was used to monitor cell growth. Cells were plated in a 96 Greiner micro clear plate and imaged every 4 hr. The default software parameters for a 96 well plate with a 10x objective were used for imaging. The IncuCyte software was used to calculate mean confluence from two non-overlapping bright phase images of each well.

The IncuCyte ZOOM instrument in combination with the Cell Player 96-well kinetic caspase-3/7 reagent (Essen Bioscience) were used to identify apoptosis by caspase 3/7 activity. The software was used to calculate mean green fluorescence from two non-overlapping fluorescent images of each well. Green fluorescent confluency was normalized to phase contrast confluency to determine apoptosis.

## Western blot analysis

Cells were harvested and subsequently lysed for 30 min in RIPA (25 mM Tris-HCl pH 7.6; 150 mM NaCl; 1% NP-40; 1% Sodiumdeoxycholate and 0.1% SDS) or ELB (150 mM NaCl; 50 mM Hepes pH7.5; 5 mM EDTA; 0.1% NP-40) containing protease inhibitors (Complete, Roche). Protein concentrations were measured using the BCA protein assay kit (Pierce).

The primary antibodies used were rabbit polyclonal phospho-Chk1 Ser317 (Bethyl), mouse mono-clonal Chk1 (G4; Santa Cruz), goat polyclonal CDK4 (C22; Santa Cruz), rabbit polyclonal p21 (C19; Santa Cruz), mouse monoclonal p27 (BD Transduction Laboratory), goat polyclonal Rb (C15; Santa Cruz); rabbit polyclonal Rbl1 (C18; Santa Cruz), mouse monoclonal Rbl2 (CAS14; Lab Vision), mouse monoclonal p53 (IMX25; monosan; for detection of mouse p53), mouse monoclonal p53 (DO-1; BD Biosciences; for detection of human p53), γ-tubulin (GTU-88; Sigma), rabbit polyclonal Cyclin D1 (Santa Cruz; H296) and goat polyclonal CDK4 (C22; Santa Cruz). Secondary antibodies used were IR Dye 800CW Goat anti-Mouse IgG, Goat anti-Rabbit IgG and Donkey anti-Goat IgG (Licor) and HRP-conjugated Goat anti-Mouse and Goat anti-Rabbit (Dako).

## Immunofluorescence

For Rad51 and γ-H2AX immunofluorescence staining, cells were cultured on cover slides, washed with PBS and fixed for 5 min using 4% paraformaldehyde (Merck). Cells were permeabilized by 0.1% Triton-X100 (sigma) in PBS for 5 min. Subsequently, cells were washed three times using staining buffer (0.15% glycine (Merck), 0.5% Bovine Serum Albumine (BSA, Sigma) in PBS) and incubated for 1 hr at room temperature in staining buffer. Cells were incubated for 4 hr and 1 hr with primary and secondary antibodies, respectively.

For CldU and γ-H2AX immunofluorescence, cells were cultured on cover slides, incubated with CldU (100 mM) for 30 min, washed with PBS and fixed for 10 min using 70% EtOH. Cells were treated with MeOH for 5 min and incubated with 1.5 M HCl for 20 min. Subsequently, cells were blocked using PBS, 0.5% Tween, 0.25% BSA, 5% FCS for 30 min. Cells were incubated with primary and secondary antibodies for 2 hr and 1 hr, respectively in PBS, 0.5% Tween,0.25 BSA. Bleaching was prevented by Vectashield (Vetcor laboratories). The primary antibodies used were rat-anti-BrdU (Clone BU1/75, Novus Biologicals), rabbit polyclonal Rad51 (a gift from Prof. Roland Kanaar) and mouse monoclonal phosphorylated H2AX (Upstate) in 1:20, 1:2500 and 1:100 dilutions, respectively. Secondary antibodies used were Alexa 488-labeled Chicken-anti-Mouse, Alexa 568-labeled Goat-anti-Rabbit and Alexa 568-labeled Goat-anti-Rat antibodies (Molecular probes) and these were used in a 1:100 dilution. DNA was stained using To-Pro3 dye (Molecular probes).

## DNA fiber analysis

Cells were pulse-labelled with 25 μM CldU followed by 250 μM IdU for 20–40 min each. After label-ling, cells were trypsinized and lysed in spreading buffer (200 mM Tris-Hcl pH 7.4, 50 mM EDTA and 0.5% SDS) before spreading on a microscope slide (Menzel-Gläser,Superfrost). Slides were fixed in methanol: acidic acid 3:1. Before immunodetection, slides were treated with 2.5 M HCl for 1 hr and 15 min. To detect CldU and IdU labelled tracts slide were incubated for 1 hr with rat-anti-Brdu (Clone BU1/75, Novus Biologicals; 1:500) and mouse-anti-BrdU (clone B44, Becton Diskinson; 1:750), respectively. Subsequently, slides were fixed with 4% paraformaldehyde for 10 min and incubated with Alexa 488-labeled goat-anti-mouse and Alexa 555-labeled goat-ant-rat (Molecular probes; 1:500) for 1 hr and 30 min. Pictures were taken with a Zeiss AxioObserver Z1 inverted microscope using a 63x lens equipped with a cooled Hamamatsu ORCA AG Black and White CCD camera and track lengths were analyzed with ImageJ software. Replication track lengths were calculated using the conversion factor 1 μm = 2.59 kb (*Jackson and Pombo, 1998*). The 1-way ANOVA (nonparametric Kruskal-Wallis test) was used for statistical analyses.

## Time-lapse microscopy

Culture dishes were transferred to a heated stage (37°C) on a Zeiss Axiovert 200M inverted micro-scope. PhC (phase contrast) images (59 ms exposure) and fluorescent images (red: 500 ms and green 300 ms exposure) were captured with a 20x/0.25 Ph1 Achroplan objective in combination with 1.6 optovar every 30 min using a cooled Hamamatsu ORCA R2 Black and White CCD-camera and appropriate filter blocks to select specific fluorescence. Images were taken in 2 × 2 binning mode (672 × 512 pixels) and processed using AxioVision Rel. 4.7.2. software.

## Flow cytometry

MEFs cultured in the presence or absence of 10% FCS were labeled with BrdU (10 μM) for 1 hr, fixed in 70% EtOH and stained with Propidium Iodide (PI). Data acquisition was performed on a Beckman

Coulter Cyan ADP and data analysis (cell cycle) was performed using FlowJo software version 7.6.1 (Tree Star, Ashland, OR, USA).

## Comet assay

Neutral comet assays were performed as described by Olive et al. (*Olive and Banáth, 2006*). Pictures of individual cells were taken with a Zeiss AxioObserver Z1 inverted microscope equipped with a cooled Hamamatsu ORCA AG Black and White CCD camera and analyzed with CASP software (http://www.casp.of.pl). The p-value was determined using 1-way ANOVA (nonparametric Kruskal-Wallis test).

## Generation of chimeric mice and teratomas

All experiments involving animals comply with local and international regulations and ethical guidelines (protocol 12026) and have been authorized by our local experimental animal committee at the Netherlands Cancer Institute (DEC-NKI). $Rb^{-/-}Rbl2^{-/-}$ ESCs were generated previously (*Dannenberg et al., 2004*). These cells were injected into C57Bl/6 blastocysts (6 cells per blastocyst) to generate chimeric mice, which were monitored weekly for retinoblastoma development. $Rb^{-/-}Rbl2^{-/-}p53^{-/-}$ ESCs were generated using CRISPR/Cas9 technology. One million cells of both cell lines were injected into the flank of Balb/c nude mice and tumors were harvested 20 days later.

## Histological and immunological analysis

Eyes and teratomas were removed immediately after euthanasia and fixed in 4% formaldehyde for at least 24 hr. For histological analysis, formaldehyde fixed tissues were embedded in paraffin, cut into 5 μm sections and stained with Hematoxilin and Eosin. The antibodies used were α-p53 (Vector-Labs), α-p-ATM (Cell signaling), α−γ−H2AX (Cell signaling), α-p-CHK2 (Cell signaling) and α-p-ATM (Genetex).

## Acknowledgements

We thank A Miyawaki for CSII-EF-MCS-mKO-hCdt1 (30/120) and CSII-EF-MCS-mAG-hGem (1/110) and R Kanaar for rabbit polyclonal Rad51 antibody. We thank L Oomen and L Brocks for help with the microscopical visualization of the DNA fibers and help with time-lapse microscopy, T Harmsen for technical support and J-Y Song for analysis of immunohistochemical tissue stainings, T van Ravesteyn, S Bakker and N Wit for fruitful discussions. This work was supported by the Dutch Cancer Society (grants NKI 2007–3790 and NKI 2014–6702) and an EMBO short-term fellowship to TvH. (194-2011).

## Additional information

### Funding

| Funder | Grant reference number | Author |
| --- | --- | --- |
| KWF Kankerbestrijding | 2007-3790 | Tanja van Harn<br>Asli Kucukosmanoglu |
| European Molecular Biology Organization | 194-2011 | Tanja van Harn |
| KWF Kankerbestrijding | 2014-6702 | Bente Benedict |

The funders had no role in study design, data collection and interpretation, or the decision to submit the work for publication.

### Author contributions

Bente Benedict, Conceptualization, Formal analysis, Validation, Investigation, Visualization, Writing—original draft, Writing—review and editing; Tanja van Harn, Conceptualization, Investigation, Writing—original draft, Writing—review and editing; Marleen Dekker, Simone Hermsen, Asli Kucukosmanoglu, Wietske Pieters, Elly Delzenne-Goette, Josephine C Dorsman, Investigation, Writing—review

and editing; Eva Petermann, Methodology, Writing—review and editing; Floris Foijer, Conceptualization, Supervision, Writing—review and editing; Hein te Riele, Conceptualization, Supervision, Funding acquisition, Visualization, Writing—original draft, Writing—review and editing

### Author ORCIDs
Bente Benedict http://orcid.org/0000-0002-7503-8527
Floris Foijer http://orcid.org/0000-0003-0989-3127
Hein te Riele http://orcid.org/0000-0003-0255-4042

### Ethics

Animal experimentation: All experiments involving animals comply with local and international regulations and ethical guidelines (protocol 12026) and have been authorized by the local experimental animal ethical committee at the Netherlands Cancer Institute (DEC-NKI).

### Decision letter and Author response
Decision letter https://doi.org/10.7554/eLife.37868.017
Author response https://doi.org/10.7554/eLife.37868.018

## Additional files

### Supplementary files
• Transparent reporting form
DOI: https://doi.org/10.7554/eLife.37868.015

### Data availability
All data generated or analysed during this study are included in the manuscript and supporting files.

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
