## [Decision Letter]

Thank you for submitting your article "Loss of p53 suppresses replication-stress-induced DNA breakage in G1/S checkpoint deficient cells" for consideration by *eLife*. Your article has been reviewed by three peer reviewers, and the evaluation has been overseen by a Guest Reviewing Editor and Jessica Tyler as the Senior Editor. The following individual involved in review of your submission has agreed to reveal his identity: David Johnson (Reviewer #2).

The reviewers have discussed the reviews with one another and the Reviewing Editor has drafted this decision to help you prepare a revised submission.

Summary:

In the presented work, the authors demonstrate the unexpected finding that inactivation of p53 in Rb triple-knockout mouse cells leads to a decrease in DNA damage, rather than an expected increase. This decrease in DNA damage correlates with increased origin firing with p53 inactivation, and allows the serum-starved cells to re-enter the cell cycle and proliferate, which could explain the frequent co-inactivation of Rb and p53 in human tumors.

All three reviewers agree that the manuscript is of potential interest. However, there are two major concerns that are necessary to be addressed before publication.

Essential revisions:

1) The biological relevance is unclear at this point as the authors use an unusual mouse system and TKO cells do not naturally occur. Therefore, a physiologically more relevant system needs to be tested. As one example, the authors could express p53 or p21 in Rb null human cancer cells and show rescue/reversion of key-conclusions of the study including origin firing, DNA damage and survival.

2) There appear to be conflicting effects of serum versus serum starvation, whereby serum starvation leads to reduced DNA damage yet increased apoptosis (Figure 1B), while the presence of serum in the same genetic system leads to increased DNA damage (Figure 3B). Similarly, it has been reported that p53 defects lead to a decrease rather than an increase in origin firing in cells irrespective of whether they are human, mouse, transformed, primary, mutant or null p53 (Roy et al., 2018). Therefore, these seeming conflicts between serum vs. non-serum conditions and whether the effect is dependent specifically on Rb or TKO need to be experimentally addressed.

---

## [Author Response]

Essential revisions:1) The biological relevance is unclear at this point as the authors use an unusual mouse system and TKO cells do not naturally occur. Therefore, a physiologically more relevant system needs to be tested. As one example, the authors could express p53 or p21 in Rb null human cancer cells and show rescue/reversion of key-conclusions of the study including origin firing, DNA damage and survival.

Indeed, cancer cells lacking all three pocket proteins (“TKO cells”) do not seem to occur naturally. However, it is known that the majority of human tumors lack the G1/S phase checkpoint and insensitivity to antigrowth signals has been identified as a hallmark of cancer. Rather than full knockout of critical effectors, abrogation of the G1/S checkpoint may also result from a combination of reduced levels of effectors (e.g. a pocket protein) and increased levels of inhibitors of G1 arrest (e.g. Cyclin D1, E2F1). Our TKO system provides a clean way to study the consequences of such a G1/S checkpoint loss, but we believe other ways of abrogating the G1/S checkpoint will have similar consequences. Indeed, we show that our findings were not dependent specifically on loss of Rb and other pocket proteins but resulted from loss of the G1/S checkpoint in general: overexpression of Cyclin D1 had similar effects as loss of the Rb family members (Figure 6). Moreover, these effects were observed in the human RPE-1 cell line, showing they were not a peculiarity of the TKO mouse embryonic fibroblast system (Figure 6).

Overexpression of Cyclin D1 is biologically relevant since comprehensive analysis of many human cancer types revealed that the gene encoding Cyclin D1 represents the second most frequently amplified locus in the human cancer genome (Beroukhim et al., 2010). In addition, in many human tumors overexpression of D type cyclins takes place in the absence of detectable genomic alterations (Hosokawa et al., 1998).

Modification: In our opinion, the experiments done in human cells and the Cyclin D experiment corroborates the physiological relevance of our findings. See Results section “Also in human cells inactivation of p53 is associated with reduced DNA DSBs”.

While the suggested experiment may indeed complement our findings, it should be noted that reintroduction of p53 is notoriously difficult as tumor cells are often adapted to abrogation of p53 activity and hence stop proliferating upon reactivation of p53.

2) There appear to be conflicting effects of serum versus serum starvation, whereby serum starvation leads to reduced DNA damage yet increased apoptosis (Figure 1B), while the presence of serum in the same genetic system leads to increased DNA damage (Figure 3B). Similarly, it has been reported that p53 defects lead to a decrease rather than an increase in origin firing in cells irrespective of whether they are human, mouse, transformed, primary, mutant or null p53 (Roy et al., 2018). Therefore, these seeming conflicts between serum vs. non-serum conditions and whether the effect is dependent specifically on Rb or TKO need to be experimentally addressed.

We do not believe there is a discrepancy between Figures 1B and 3B. It is true that p53KO reduced DNA damage in TKO-Bcl2 cells upon serum starvation (Figure 3B), while this appeared to be accompanied with higher levels of cell death (Figure 1B). However, the very low level of cell death in TKO-Bcl2 cells despite the high level of damage is caused by the fact that these cells do not proliferate but become arrested in a G2-like state (Figure 1C). Apparently, the proliferation of TKO-Bcl2-p53KO cells in the absence of serum comes at the price of low levels of (p53-independent) cell death, whereas TKO-Bcl2 cells respond to serum deprivation by cell cycle arrest rendering them insensitive to apoptotic cell death.

Furthermore, we observed that inactivation of p53 and p21^Cip1^ in TKO-Bcl2 MEFs slightly increased DNA damage in the presence of serum compared to TKO-Bcl2 MEFs. We hypothesized this was due to the regulatory function of p53 in the DNA damage response; upon inactivation of p53 or p21^Cip1^, cells tolerate a higher level of DNA damage irrespective of the culture conditions. The critical point though we wanted to make here is that serum deprivation did *not* increase DNA damage in TKO-Bcl2-p53KO or TKO-Bcl2-p21KO MEFs while a clear increase in DNA damage was seen in TKO-Bcl2 MEFs.

Modification: We have now better explained this observation in the subsection “p53/p21^Cip1^ knockout suppresses DSBs formation”, stressing the difference between basal levels of DSBs and induction of DSBs by mitogen deprivation.

We subsequently showed that reduced induction of DNA damage in serum-starved TKO-Bcl2-p53KO cells was accompanied by increased origin firing. Indeed, this may conflict with the recent publication of Roy et al. (2018). Roy et al. identified a novel role for p53 in balancing replication homeostasis. In wild type cells experiencing replication stress, p53 binds to replication forks and facilitates replication fork restart by recruitment of MLL3 and MRE11. Upon knockout or mutation of p53, MLL3 and MRE11 are not recruited and mutagenic RAD52 and POLθ pathways can act at reversed forks and induce genomic instability. In addition, the authors report a clear decrease inthe level of origin firing in p53 KO MEFs compared to wild type MEFs in the presence of a low dose of hydroxyurea (HU).

At first glance, the latter indeed seems opposite to our findings. By comparing the observations and experimental setups, we envisage several explanations for this discrepancy.

1) Roy et al. clearly showed that the new function of p53 in regulating DNA homeostasis is transcription-independent, whereas we found that the role of p53 in suppressing origin firing is dependent on its downstream transcription factor p21^Cip1^. The two effects therefore rely on different functions of p53 that may operate side by side: it doesn’t seem without logic that upon replication stress, p53 operates twofold to properly process stalled replication forks and to suppress new origin firing.

2) Nevertheless, Roy et al. clearly observed reduced origin firing upon ablation of p53. In their experiments they apply first color labeling (green) of nascent DNA followed by second color labeling (red) in the presence of HU, that is continuous replication stress. Their subsequent experiments make a strong case for p53 to promote replication fork restart. Indeed in the absence of p53, the level of stalled forks, that is green only fibers, increase. Apparently many of the green fibers become immediately stalled upon HU treatment, that is without a trace of red, and cannot resume without p53. Would it be possible then that many newly initiated forks become immediately stalled and cannot resume, and hence remain invisible as a red fiber? However, a third alternative seems more likely:

3) The differential effect of p53 ablation on origin firing may also be related to the level of replication stress induced. Whereas Roy et al. used low doses of HU to induce replication stress, we cultured cells in the absence of mitogens, a condition inducing stringent replication stress observed by the drastic reduction of the replication fork speed. As suggested by Roy et al., low levels of replication stress induced replication fork stalling, but not replication fork collapse, that is DNA DSBs. In contrast, high levels of replication stress induced DSB formation. Thus, at low doses of HU, replication forks stall and p53 operates to protect and properly restart these forks. At high levels of replication stress, stalled replication forks get broken and activate p53 to suppress new origin firing. Hence, ablation of p53 increases new origin firing, which may help to rescue stalled forks preventing further breakage.

To study this hypothesis, we exposed cells to a low dose of HU (300 µM; similar to the dose used by Roy et al.) and found no induction of DNA DSBs. In contrast, high levels of HU (2 mM) did induce DNA DSBs (see Figure 5—figure supplement 2A). Furthermore, consistent with Roy et al., p53 KO reduced origin firing after exposure to la low dose of HU, while at a high dose of HU, the level of origin firing was not changed upon loss of p53. (See Figure 5—figure supplement 2B). Based on this observation, we believe the severity of replication stress dictates the outcome of p53 ablation.

Modification: We describe these new findings in the revised manuscript. See the Results section “Reduced DNA breakage by p53 knockout is associated with increased origin firing”, the Discussion, fourth paragraph, and Figure 5—figure supplement 2.